# Functional and diffusion MRI reveal the neurophysiological basis of neonates' noxious-stimulus evoked brain activity

Luke Baxter [1], Fiona Moultrie[1], Sean Fitzgibbon[2], Marianne Aspbury [1], Roshni Mansfield[1], Matteo Bastiani[2,3,4], Richard Rogers[5], Saad Jbabdi[2], Eugene Duff [1,2,6] & Rebeccah Slater [1,6✉]

Understanding the neurophysiology underlying neonatal responses to noxious stimulation is central to improving early life pain management. In this neonatal multimodal MRI study, we use resting-state and diffusion MRI to investigate inter-individual variability in noxious-stimulus evoked brain activity. We observe that cerebral haemodynamic responses to experimental noxious stimulation can be predicted from separately acquired resting-state brain activity ($n = 18$). Applying this prediction model to independent Developing Human Connectome Project data ($n = 215$), we identify negative associations between predicted noxious-stimulus evoked responses and white matter mean diffusivity. These associations are subsequently confirmed in the original noxious stimulation paradigm dataset, validating the prediction model. Here, we observe that noxious-stimulus evoked brain activity in healthy neonates is coupled to resting-state activity and white matter microstructure, that neural features can be used to predict responses to noxious stimulation, and that the dHCP dataset could be utilised for future exploratory research of early life pain system neurophysiology.

[1] Department of Paediatrics, University of Oxford, Oxford, UK. [2] FMRIB, Wellcome Centre for Integrative Neuroimaging, University of Oxford, Oxford, UK. [3] Sir Peter Mansfield Imaging Centre, School of Medicine, University of Nottingham, Nottingham, UK. [4] NIHR Biomedical Research Centre, University of Nottingham, Nottingham, UK. [5] Nuffield Department of Anaesthetics, John Radcliffe Hospital, Oxford, UK. [6]These authors contributed equally: Eugene Duff, Rebeccah Slater. ✉email: rebeccah.slater@paediatrics.ox.ac.uk

Neonates routinely undergo numerous painful procedures as part of standard clinical care shortly after birth during their stay in hospital[1]. Their lack of verbal communication, brief extra-uterine medical history, and ambiguity in behavioural and physiological responses that underpin infant pain scales[2], lead to a high degree of uncertainty in clinical decision-making related to the treatment of neonatal pain. Understanding and anticipating an individual newborn's response to noxious input would advance efforts of personalised pre-emptive pain minimisation in this vulnerable population.

In experimental settings, a multitude of complementary behavioural, physiological, and neural measures are used in an attempt to quantify neonatal pain, with a high degree of individual variability observed within all modalities[3–6]. Due to the neural origin of pain and the recent feasibility of collecting multiple high-quality MRI imaging modalities of the neonatal brain within a single scan session, we used a multimodal MRI approach to investigate the neurophysiological basis for individual variability in neonates' blood oxygen level-dependent (BOLD) responses to noxious stimulation. Our noxious stimulation paradigm involved applying a mild experimental sharp-touch stimulus (PinPrick Stimulator, MRC Systems) to the neonate's foot, evoking brain activity known to be similarly evoked by a range of tissue-damaging medical procedures, such as blood sampling, vaccinations, and cannulations[3,7,8]. The pinprick stimulus produces responses of lower amplitude than those of clinical procedures and does not cause behavioural distress[3], but activates A-fibre nociceptors in the periphery[9] and elicits noxious-evoked brain activity in the cerebral cortex[10–13], making it a useful experimental tool to better understand neonatal pain processing.

Due to the emergent and multifaceted nature of pain[14,15], we focus on assessing the overall BOLD response amplitude. While this BOLD response neither directly reflects nociception, the neural process of encoding noxious stimuli[16], nor pain perception, the unpleasant sensory and emotional subjective experience[16], it is a pertinent and accessible feature of central importance to understanding neonates' neural responses to noxious input and the neurophysiology of the early developing pain system. This overall noxious-evoked BOLD response pattern resembles that of adults[10] and expresses the adult Neurologic Pain Signature (NPS)[17], a multivariate fMRI signature predictive of adult verbal reports of physical pain. The overall response captures inter-individual variability in the multidimensional noxious-evoked activity, which is linked to the pre-stimulation functional status of the descending pain modulatory system[18] and is likely to be driven by variability in sensory-discriminative, cognitive, and emotional aspects.

Furthermore, a holistic multidimensional noxious-evoked brain response metric reflects and should facilitate future harmonisation with existing validated multidimensional infant clinical pain assessment tools, such as the widely used PIPP-R scale (Premature Infant Pain Profile Revised)[19–21], which integrates across multiple pain-relevant behavioural and autonomic response features to provide a reliable overall measure of this complex phenomenon[22]. Due to the subjective nature of pain and non-verbal nature of neonates, having to rely on objectively measured noxious-evoked response features for infant pain measurement is a major challenge for the field of infant pain research, and thus facilitating this cross-modality integration is vital for mitigating limitations of each individual objective assessment approach[23,24]. We augment the analysis of overall BOLD responses with a parallel study of the expression in this data of Neurosynth-derived templates[25] and adult pain signatures, providing insight into the processes contributing to the observed responses.

To better understand the neurophysiological basis for individual variability in neonates' overall BOLD responses to noxious stimulation, we test whether noxious-evoked responses can be predicted from nociception-free resting-state brain activity. To determine whether response amplitudes reflect the current state of the infant, or a developmental trait effect, we further test whether responses are associated with underlying white matter microstructure. We mitigate the small sample size limitation inherent to neonatal fMRI pain studies by identifying consistent findings in a large independent age-matched sample from the Developing Human Connectome Project (dHCP) dataset (http://www.developingconnectome.org).

A high degree of correspondence between resting-state and task-related brain activities has been observed in adult fMRI studies[26,27]. In adults, fMRI-recorded resting-state activity is a distinguishing feature of an individual's brain functionality[28], predicts individuals' task-related brain activity under both experimental[29] and clinical conditions[30], as well as adults' individual pain sensitivities[31]. While analogous studies have not been conducted in neonatal populations, large-scale resting-state networks (RSNs) are detectable using fMRI from birth and correspond to adult canonical resting-state and task-response networks[32,33], suggesting a similar functional coupling could exist at this early developmental stage.

Previous studies have demonstrated the sensitivity of neonates' noxious-evoked cerebral activity to sleep state[13] and physiological stress[34]. To disambiguate temporally stable trait effects from transient state effects, which are arguably of higher relevance for clinical pre-emptive decision making, we assess associations between neonates' noxious-evoked response amplitudes and underlying white matter microstructure using diffusion MRI (dMRI) data. These microstructural features reflect the integrity of developing structural connectivity, constraining noxious-evoked responses. Temporal stability was assessed through association with microstructure rather than looking at stability across multiple test occasions, as neonates could only be tested on a single occasion.

Due to the large number of potential white matter features that could be studied and the lack of neonatal research into associations between noxious-evoked activity and white matter microstructure that could guide the feature selection process, we adopted a two-part two-dataset approach, involving exploration of a range of structure-function associations in the dHCP dataset to formulate data-driven hypotheses, followed by independent confirmation of these hypotheses in our noxious stimulation paradigm dataset. We use the dHCP dataset to explore possible structure-function relationships due to its larger sample size and thus greater statistical power. Given that the dHCP dataset does not include noxious stimulation paradigm data, we generate a predicted noxious-evoked response amplitude per neonate from their resting-state data using the prediction model originally trained in our local noxious stimulation paradigm dataset. We focus on 16 white matter tracts previously used in a recent dHCP dMRI publication[35] and three tensor model parameters generated by the dHCP dMRI preprocessing pipeline[35]: mean diffusivity (MD), fractional anisotropy (FA), and mean kurtosis (MK). Structure-function associations identified in this exploratory analysis were subsequently tested in our noxious stimulation paradigm dataset, for which measured noxious-evoked response amplitudes are available.

This work provides insight into the neurophysiological basis for normative variability in cerebral responses to noxious input in healthy neonates. We demonstrate a pain-relevant neural structure-function relationship, and the observed coupling between resting-state and noxious-evoked response activities provides proof-of-concept that neonates' nociception-free resting-state brain activity can be used to predict their brain response to noxious stimulation.

## Results

**Neonates displayed wide variability in haemodynamic response amplitude to noxious stimulation.** We quantified the change in brain activity evoked by a 128 mN pinprick experimental noxious stimulus to the foot in 18 healthy neonates (Fig. 1). The noxious-evoked response was localised to functional brain regions classically considered part of the adult nociceptive pain system, including pre- and post-central gyri, opercular and insular cortices, and the thalamus[36] (Fig. 2a). Responses were highly variable between subjects including positive, negligible, and negative BOLD responses (Fig. 1 heat maps). Summarising each neonate's noxious-evoked response map relative to the group average response map, response amplitudes ranged from −0.87 to 5.60 (Fig. 1 scalar values).

For each neonate, the noxious-evoked response was well fit by the term-neonate double gamma haemodynamic response function (HRF). There were no obvious signs of gross artefactual errors such as head motion-related spikes (Supplementary Fig. 1) and no association with variable response latencies assessed by HRF goodness-of-fit (Supplementary Fig. 3), suggesting that the estimated noxious-evoked activity reflects physiologically meaningful cerebral BOLD responses to noxious input.

To gain insight into processes underlying the observed noxious-evoked activity, we assessed the expression of two template maps that have been independently linked to pain in adults. Both the Neurosynth[25] pain association test map (https://www.neurosynth.org/analyses/terms/pain), derived from a search of a meta-analytic database of fMRI study activation co-ordinates for the keyword "pain", and the adult NPS[37], which predicts variations in adult verbal reports of pain intensity, were significantly expressed (Fig. 2bi and Table 1 T-test results). We also assessed the expression of a number of related and control templates. As negative controls, we assessed the Neurosynth "visual" pattern as well as the adult social rejection pain signature[37], and found neither to be expressed. Of the sensory, cognitive, and emotional Neurosynth patterns assessed, only the "nociceptive" pattern was significantly expressed, while patterns for "attention", "unpleasant", "salience", and "arousal" were not (Fig. 2bi and Table 1 T-test results).

In addition to group average template expression, we assessed associations between the inter-individual variability in overall noxious-evoked response amplitudes (regression coefficients) and the correspondence between neonates' noxious-evoked response maps and each template (correlation coefficients). Of the nine adult templates assessed, significant associations were only observed between noxious-evoked response amplitudes and adult template expression for NPS and Neurosynth Pain and Nociceptive templates (Fig. 2ii–iii and Table 1 Correlation results). Thus, the larger the overall BOLD response amplitude to noxious stimulation, the closer the correspondence with adult pain and nociceptive signatures. Correlation results for all templates are presented in Supplementary Fig. 9 and Table 1. Collectively, these adult template results support the interpretation that the neonates' overall noxious-evoked responses are pain-relevant signals.

**Nine RSNs were replicable across the noxious stimulation paradigm and dHCP datasets.** In the same cohort ($n = 18$), nine RSNs were robustly identified from separate resting-state scans using probabilistic functional mode (PFM) analysis (Fig. 3). These included six sensory and motor networks (two visual, two auditory, and two somatomotor networks (SMN)) and three cognitive networks (default mode, dorsal attention, and executive control networks). To consider a network robust and suitable for inclusion in subsequent analyses, networks needed to be replicable across both the noxious stimulation paradigm dataset ($n = 18$) and an independent age-matched dHCP dataset ($n = 242$) previously analysed using PFM[38]. Eleven non-zero modes of variation were identified in the noxious stimulation paradigm dataset, nine of which corresponded to networks in the dHCP dataset, assessed using spatial correlation followed by visual confirmation (Fig. 3). Matched networks were highly consistent between datasets with spatial correlations between unthresholded maps ranging from 0.62 to 0.89 (mean = 0.77) (Fig. 3 scalar values).

In our local dataset, RSN amplitudes were quantified using multiple regression of the nine dHCP networks (Fig. 3 bottom row) onto each neonate's resting-state data, and each resulting network timeseries summarised as an amplitude using median absolute deviation (MAD), which ensures robustness to outliers. Examining network timeseries, there were no obvious signs of gross artefactual errors (Supplementary Fig. 2). While network timeseries outliers existed, they have minimal influence on

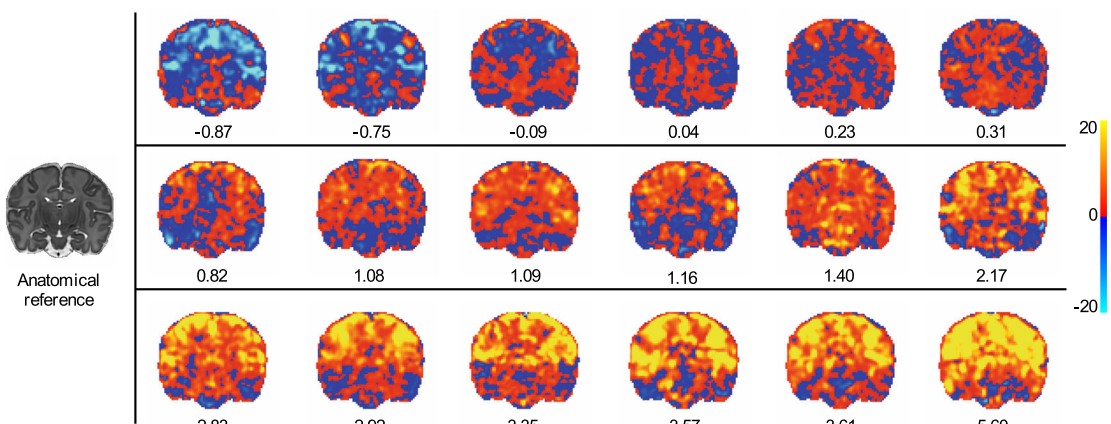

**Fig. 1 Noxious-evoked response amplitudes.** A noxious-evoked response BOLD activity map is presented for each neonate ($n = 18$) and ordered according to the overall response amplitude. The maps are general linear model regression parameter maps (regression parameters are scaled according to colour bar). The anatomical reference (left) provides structural detail for orientation. All maps are displayed at this slice position to maximally emphasise the range of individual variability in response amplitudes. Unthresholded maps are used for visualisation to demonstrate the range of evoked response amplitudes from negative to negligible to positive amplitudes, without introducing the issues inherent to the application of arbitrary thresholds. The scalar value presented below each map is a summary measure that represents the overall noxious-evoked response amplitude relative to the group average. Source data are provided as a Source Data file.

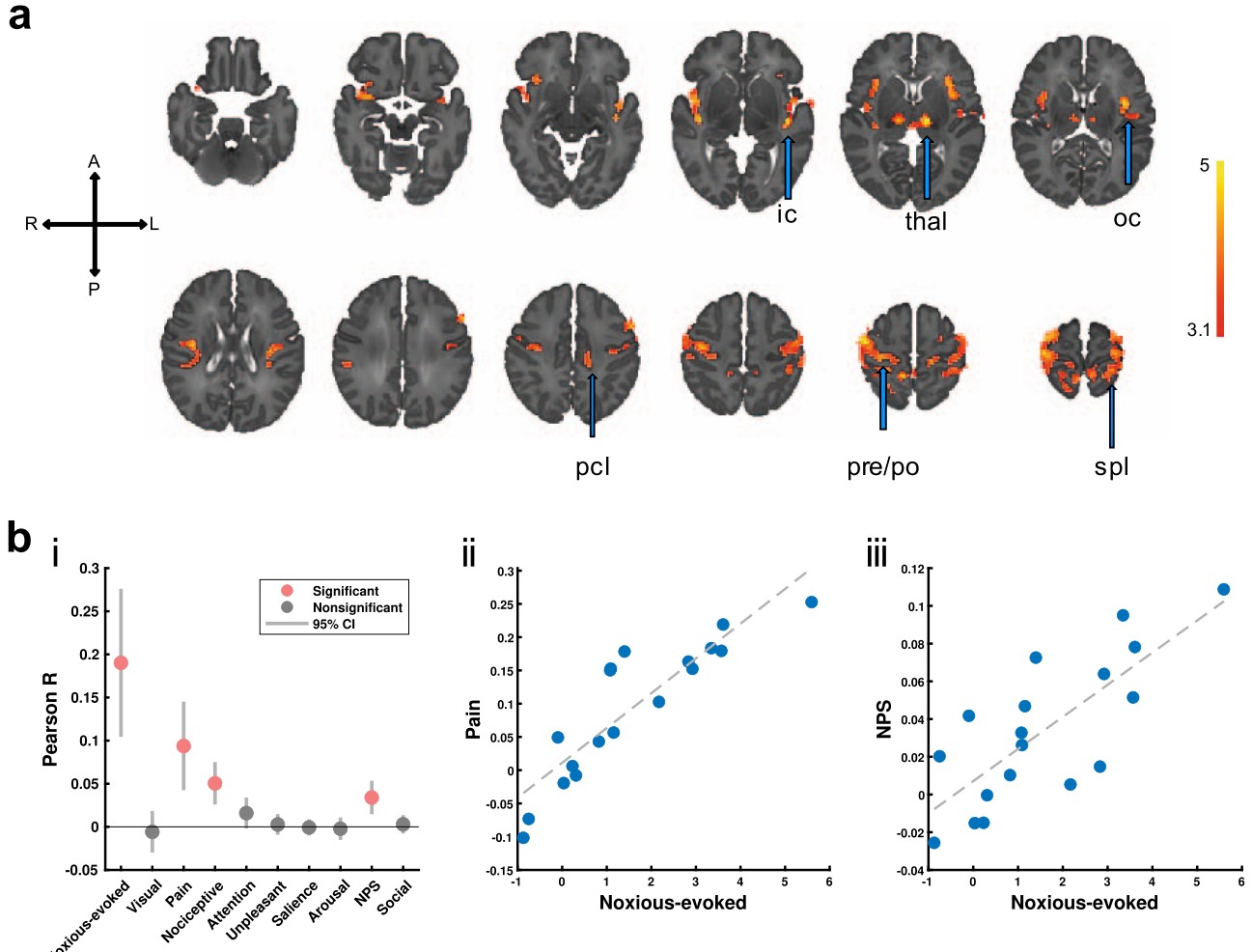

**Fig. 2 Noxious-evoked responses are pain-relevant signals. a** The thresholded group average noxious-evoked map displays t-statistics in statistically significant clusters (t-statistics are scaled according to colour bar). Activity is localised to regions classically considered part of the adult nociceptive pain system, including the pre- and post-central gyri (pre/po), paracentral and superior parietal lobules (pcl and spl), opercular and insular cortices (oc and ic), and thalamus (thal). A, P, L, R = anterior, posterior, left, right. **bi** For each infant, expression of functional templates (*x*-axis) is assessed as whole-brain Pearson correlations between the template and neonates' noxious-evoked response maps (*y*-axis). Group average template expression was assessed using two-tailed *t*-tests (*n* = 18). Grey and red dots represent the group mean correlation coefficient, with grey bars displaying 95% confidence intervals (CI). The templates used included one map derived from the current neonatal dataset (Noxious-evoked), seven Neurosynth maps (Visual to Arousal), and two pain subtype signature maps (NPS and Social). The thresholded noxious-evoked map (displayed in part a) is a positive control. Visual is the Neurosynth negative control, and Social Rejection Pain is the pain signature negative control. The Neurologic Pain Signature (NPS) and Neurosynth Pain and Nociceptive templates were significantly expressed in this group of neonates, while none of the negative controls or other Neurosynth templates were significantly expressed. **ii-iii** Using two-tailed Pearson correlation tests to assess inter-subject variability in noxious-evoked responses, associations exist between the overall noxious-evoked response amplitudes (regression parameters) and both NPS and Neurosynth Pain correspondences (correlation coefficients): NPS Pearson *r* = 0.77 (*p* = 0.0002), Neurosynth Pain Pearson *r* = 0.89 (*p* = 0.0001). The dashed grey line is the least squares fit. The stronger the neonatal BOLD response amplitude to the noxious stimulus, the closer the correspondence with both adult pain signatures. *T*-test and correlation test results for all templates are summarised in Table 1, and correlation plots for all Neurosynth and pain signature templates are displayed in Supplementary Fig. 9. Source data are provided as a Source Data file.

amplitudes estimated using MAD. Finally, no association existed between neonates' RSN timeseries outlier content and noxious-evoked response amplitudes (Supplementary Fig. 3), suggesting that individual variability in response amplitudes was not associated with individual variability in RSN timeseries quality.

**RSN amplitudes predicted noxious-evoked response amplitudes.** We predicted neonates' overall noxious-evoked response amplitudes (Fig. 1 scalar values) from their nociception-free RSN amplitudes with statistically significant accuracy ($R^2 = 0.62$, $p = 0.0005$) (Fig. 4; Table 2). RSN amplitudes were also predictive

of both the Neurosynth ($R^2 = 0.46$, $p = 0.0066$) and NPS ($R^2 = 0.42$, $p = 0.0131$) response magnitudes (Table 2). Using a linear support vector regression (SVR) model, predictions were generated using leave-one-out cross-validation, including cross-validated adjustment for several confounds (see "Methods").

Three resting-state imaging confounds (head motion, cerebrospinal fluid and white matter amplitudes), were additionally tested in a multivariate model but were not predictive of the neonates' noxious-evoked response amplitudes (Fig. 4; Table 2). Similarly, six non-fMRI clinical variables, which included postmenstrual age (PMA), gestational age (GA), postnatal age (PNA), birth weight (BW), total brain volume (TBV), and sex,

**Table 1 Noxious-evoked responses are pain-relevant signals.**

|  | Noxious-evoked | Visual | Pain | Nociceptive | Attention |
|---|---|---|---|---|---|
| T-test | 4.68[a] (0.0007) | −0.51 (0.62) | 3.85[a] (0.0022) | 4.34[a] (0.0013) | 1.88 (0.078) |
| Correlation | 0.88[a] (0.0001) | 0.18 (0.47) | 0.89[a] (0.0001) | 0.87[a] (0.0001) | 0.28 (0.26) |

|  | Unpleasant | Salience | Arousal | NPS | Social |
|---|---|---|---|---|---|
| T-test | 0.49 (0.66) | −0.15 (0.89) | −0.34 (0.76) | 3.71[a] (0.0016) | 0.58 (0.56) |
| Correlation | −0.33 (0.18) | −0.11 (0.66) | −0.32 (0.19) | 0.77[a] (0.0002) | 0.36 (0.14) |

T-test results assess the group average presence or absence of template expression within the neonates' ($n = 18$) noxious-evoked response maps (see Fig. 2bi). T-statistics and p-values are presented for each template. Correlation results assess the correspondence between the inter-individual variability in overall noxious-evoked response amplitudes and the correspondence between neonates' noxious-evoked response maps and each template (see Supplementary Fig. 9). Pearson correlation coefficients and p-values are presented for each template. Two-sided uncorrected p-values are presented in parentheses.
[a]Statistically significant using significance level Bonferroni-corrected for effective number of tests[101,102] ($\alpha = 0.05/ 8.3403 = 0.0060$). Source data are provided as a Source Data file.

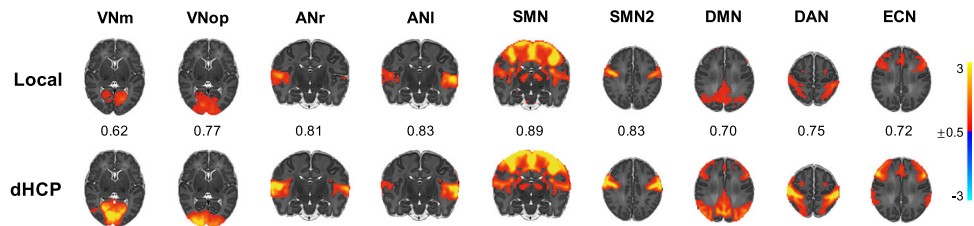

**Fig. 3 Nine resting-state networks replicated across two independent datasets.** Each resting-state network map is a thresholded group-level probabilistic functional mode (PFM) map identified in the locally collected noxious stimulation paradigm dataset ($n = 18$ subjects' resting-state data) (top row, Local) and the age-matched dHCP dataset ($n = 242$ subjects' resting-state data) (bottom row, dHCP). These PFM posterior probability maps are thresholded to highlight qualitative correspondence (means of posterior distributions are scaled according to colour bar). The scalar value shown between matched maps is the spatial Pearson correlation coefficient between unthresholded maps highlighting quantitative correspondence. VNm medial visual network, VNop occipital pole visual network, ANr right auditory network, ANl left auditory network, SMN somatomotor network, DMN default mode network, DAN dorsal attention network, ECN executive control network. Source data are provided as a Source Data file.

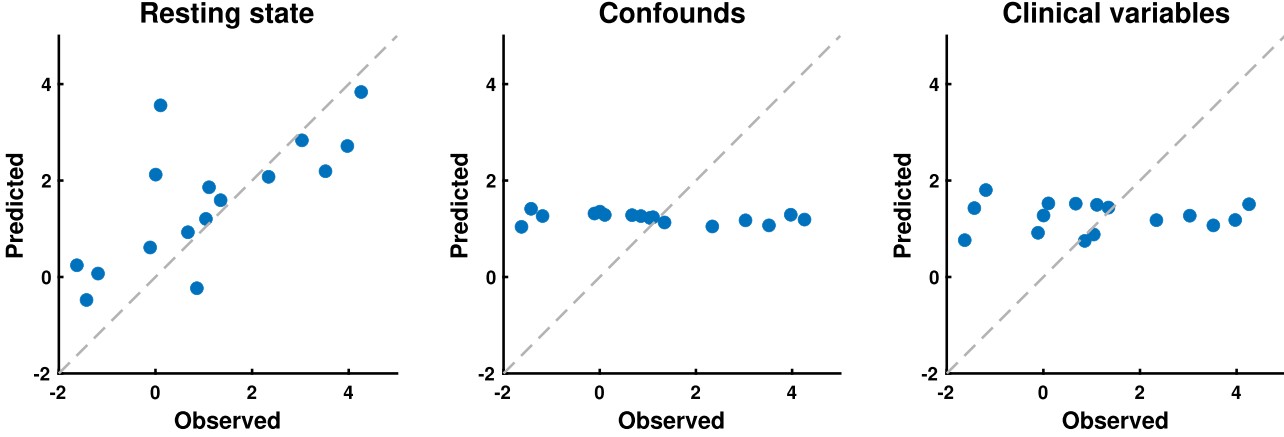

**Fig. 4 Predicting noxious-evoked response amplitudes from non-noxious data.** For all plots, each blue dot represents an out-of-sample cross-validated prediction for a single neonate ($n = 18$), and the dashed grey line is the $y = x$ line along which perfect predictions would lie. The x-axis is the observed noxious-evoked response amplitude (after cross-validated confound regression), and the y-axis is the predicted noxious-evoked response amplitude. Predictions were generated based on three sets of predictors: (left) the resting-state network amplitudes; (middle) resting-state imaging confounds, which included head motion, CSF amplitude, and white matter amplitude; and (right) clinical variables, which included age (gestational, postmenstrual, and postnatal), birth weight, total brain volume, and sex. Source data are provided as a Source Data file.

were tested in a multivariate model and were also not predictive (Fig. 4; Table 2). The lack of association between noxious-evoked response amplitudes and resting-state imaging confounds suggest that the predictive value of RSN amplitudes was not mediated by undesirable confounding features of resting-state data, but rather by the correspondence between resting-state and noxious-evoked brain activities. These brain function similarities could not be explained by the biologically interesting variables of age, BW, brain volume, or sex.

Examining the correlation polarity of each network individually, eight of nine networks were positively correlated with noxious-evoked response amplitudes, with the single negative correlation being negligible (executive control network, $r = -0.03$) (Supplementary Fig. 4a). Examining the predictive value of each network individually, only the SMN exhibit notable predictive value, with a prediction performance ($R^2 = 0.60$) comparable to our multivariate prediction model performance ($R^2 = 0.62$) (Supplementary Fig. 4b). This suggests a near-global positive

**Table 2 Noxious-evoked response amplitude prediction performance.**

| Predictors | Responses | $R^2$ | RMSE | $R_{Sp}$ |
|---|---|---|---|---|
| Resting state | Overall | 0.62[a] (0.0005) | 1.57[a] (0.0005) | 0.79[a] (0.0013) |
| | Neurosynth pain | 0.46[a] (0.0066) | 0.15[a] (0.0066) | 0.65[a] (0.0109) |
| | NPS | 0.42[a] (0.0131) | 0.025[a] (0.0131) | 0.62 (0.0186) |
| Clinical variables | Overall | 0.11 (0.2615) | 2.42 (0.2615) | 0.19 (0.3661) |
| Confounds | Overall | 0.081 (0.5518) | 2.46 (0.5518) | 0.14 (0.4452) |

Each row contains results for a specific set of predictors and responses. Each results column contains a prediction performance metric.
$R^2$ coefficient of determination (sums-of-squares formulation), RMSE root mean squared error, $R_{Sp}$ Spearman's rank correlation coefficient.
[a]Statistically significant using significance level Bonferroni-corrected for effective number of tests[101,102] ($\alpha = 0.05/3.644 = 0.0137$).

association between RSN amplitudes and noxious-evoked response amplitudes, with meaningful network-to-network variability in association strength such that the most robust functional coupling was with the functionally relevant SMN. These univariate correlation and prediction analyses did not reveal any individual resting-state imaging confound or clinical variable to have an association with noxious-evoked response amplitudes (Supplementary Fig. 4).

**Noxious-evoked response amplitudes were significantly negatively associated with white matter mean diffusivity.** The SVR prediction model was trained on neonates in the noxious stimulation paradigm dataset ($n = 18$) to map from confound-adjusted RSN amplitudes to confound-adjusted noxious-evoked response amplitudes. Using this model, predicted noxious-evoked response amplitudes were generated for an age-matched dHCP sample ($n = 215$) for which both resting-state and diffusion data were available. These predicted noxious-evoked response amplitudes were used for the structure-function exploratory arm analyses due to the large sample size, and the results formed the basis for data-driven hypotheses regarding noxious stimulation-related structure-function associations. These hypotheses were subsequently tested in the structure-function confirmatory arm analyses using the local noxious stimulation paradigm dataset, for which measured noxious-evoked response amplitudes were available.

In our exploratory arm, we assessed MD, FA, and MK across 16 bilateral white matter tracts[35]. Of the 48 univariate correlations tested (16 tracts × 3 parameters), the predicted noxious-evoked response amplitudes were significantly negatively correlated with MD in five white matter tracts: anterior and superior thalamic radiation, corticospinal tract, forceps minor, and uncinate fasciculus (Fig. 5). Negative associations with MD and positive associations with FA existed for all tracts. However, we limited our data-driven hypotheses to the MD of these five specific tracts, because they were the most robust subset of structure-function associations, and in addition to global effects, we expect some specificity to certain functionally relevant tracts.

Due to the consistent negative correlation polarity, the variance of these five tracts was pooled using principal component analysis (PCA). The first principal component of MD across these five tracts (MD PC1) accounted for 83.6% of cross-subject variance, and as expected, was negatively correlated with the predicted noxious-evoked response amplitudes: $r = -0.25$, $p = 0.0001$ (this statistical test is biased due to circularity in explanatory variable selection[39], but the bias is restricted to the exploratory arm) (Fig. 6a). These significant negative correlations between noxious-evoked response amplitudes and MD formed the basis for two testable hypotheses: (i) the coefficient polarities for correlations between noxious-evoked response amplitudes and MD are negative for each of these five tracts, and (ii) MD PC1 across these five tracts is significantly negatively correlated with noxious-evoked response amplitudes.

In our confirmatory arm, to validate these exploratory findings and the resting-state prediction model underpinning them, we tested whether measured noxious-evoked brain activity in the noxious stimulation paradigm dataset ($n = 17$) was also dependent on the same structural brain properties. For each of the five white matter tracts, MD was negatively correlated with noxious-evoked response amplitudes (Fig. 6b). In addition, MD PC1 accounted for 88.82% of the between-subject variance and was significantly negatively correlated with neonates' noxious-evoked response amplitudes: $r = -0.454$, $p = 0.034$ (Fig. 6b). Thus, within our noxious stimulation paradigm dataset, over 20% of the between-subject variation in noxious-evoked response amplitudes could be explained by the MD of these five functionally relevant white matter tracts.

We also observed widespread negative associations with MD and positive association with FA (Supplementary Fig. 6a), reproducing the global structure-function association effects observed in the dHCP dataset (Fig. 5). In addition, there was noticeable tract-to-tract variability in association strength in both datasets. Comparing this tract-to-tract variability between datasets, the high similarity was observed for MD ($r = 0.62$) (Supplementary Fig. 6b), suggesting that tracts with strongest associations between MD and noxious-evoked response amplitudes were relatively consistent between datasets. We compared the variance in noxious-evoked response amplitudes explained by MD PC1 of these five specific tracts to that of the global signal (MD PC1 across all 16 tracts) as well as to all possible combinations of the 16 tracts (Supplementary Fig. 7). The specific 5-tract MD PC1 $r^2 = 0.21$ outperformed the global 16-tract MD PC1 $r^2 = 0.17$, and featured in the 92.3th percentile of the distribution of all possible tract combinations.

These results suggest global structure-function associations exist between noxious-evoked response amplitudes and white matter MD and FA. Meaningful tract-to-tract variability in MD association was reproducible between datasets, such that a subset of pain-relevant tracts exhibited greater explanatory value than a global MD metric, which does not account for tract-to-tract variability in functional relevance.

## Discussion

This multimodal MRI study demonstrates that individual variability in neonatal noxious-evoked brain activity is dependent on both structural and functional architectures of the brain. Using resting-state fMRI and white matter dMRI, we show that both local and global features of resting-state activity and white matter microstructure provide insight into the neurophysiological basis for neonatal cerebral responses to noxious stimulation. We train a cross-validated prediction model to map from RSN amplitudes to noxious-evoked response amplitudes, and demonstrate that the model generalises well to an independent age-matched sample from the dHCP. This demonstrates that neonates' noxious-evoked response amplitudes can be predicted from their nociception-free resting-state activity, and opens the possibility of

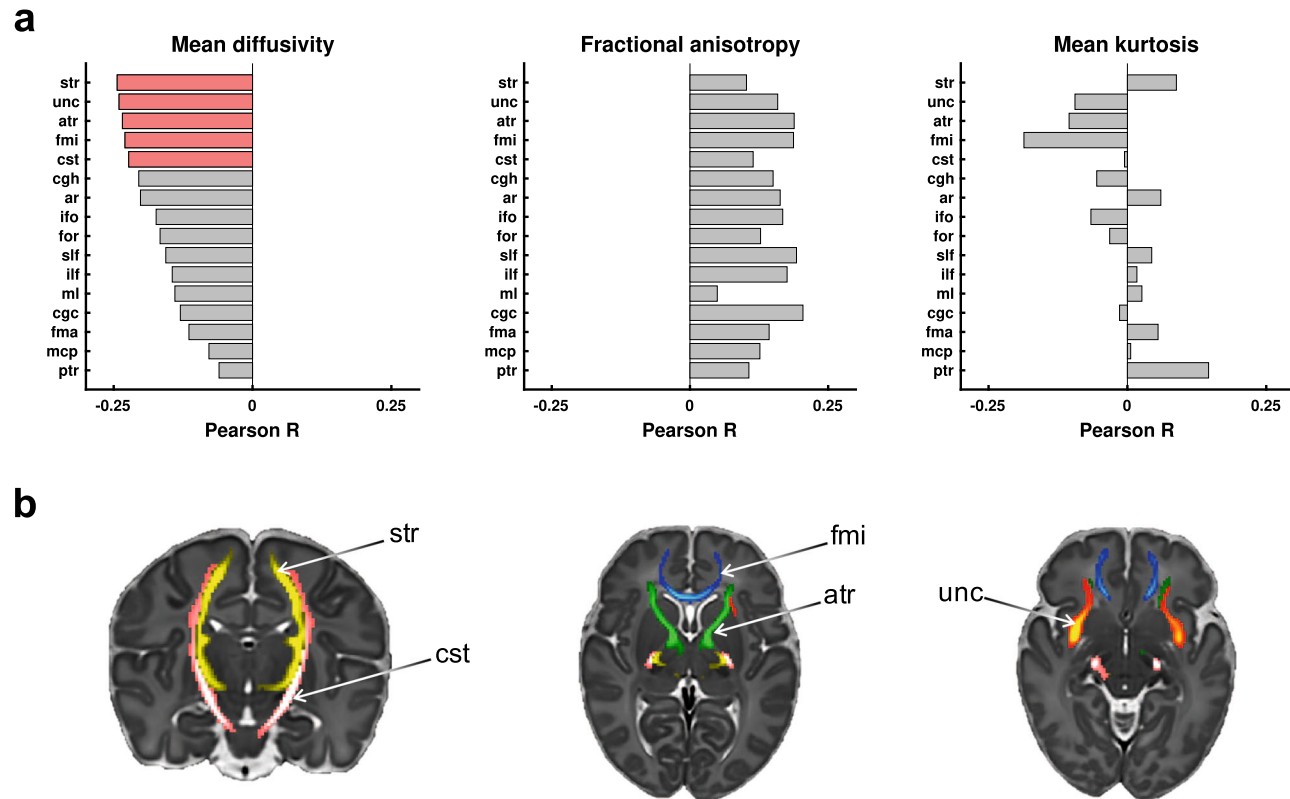

**Fig. 5 Exploration of structure-function associations in the dHCP dataset.** The structural feature per white matter tract is the voxelwise mean diffusion parameter: mean diffusivity, fractional anisotropy, and mean kurtosis. The functional feature is the predicted noxious-evoked response amplitude, generated using the resting-state prediction model. **a** The three plots display the Pearson correlation coefficients (x-axis) between response amplitudes and diffusion parameters for all 16 white matter tracts (y-axis). The white matter tracts are ordered according to the mean diffusivity correlation coefficients for which statistically significant results were found (red). Statistical significance is FWER-corrected for multiple testing across all 48 Pearson correlation tests. **b** Maps displaying the five bilateral white matter tracts statistically significantly related to predicted noxious-evoked response amplitudes. ar acoustic radiation, atr anterior thalamic radiation, cgc cingulate gyrus part of the cingulum, cgh parahippocampal part of the cingulum, cst corticospinal tract, fma forceps major, fmi forceps minor, for fornix, ifo inferior fronto-occipital fasciculus, ilf inferior longitudinal fasciculus, mcp middle cerebellar peduncle, ml medial lemniscus, ptr posterior thalamic radiation, slf superior longitudinal fasciculus, str superior thalamic radiation, unc uncinate fasciculus. Source data are provided as a Source Data file.

performing future exploratory noxious stimulation paradigm research in the large publicly available dHCP dataset, for which noxious-evoked response data have not been collected.

Using the 128 mN pinprick stimulation paradigm, the network of brain regions responding to noxious input in neonates has previously been shown to closely match that in adults[10]. Here, we used a group-average response template to assess individual variability in evoked amplitudes. We confirmed the central importance of multiple brain regions classically considered part of the adult nociceptive pain system[36], and demonstrated expression of a Neurosynth pain pattern and the NPS, two activity patterns associated with adult pain.

Pain is a multidimensional phenomenon that transcends nociception with sensory, cognitive, and emotional components to response variability[14,15]. The observed inter-subject variability in response amplitudes assessed in the current study may be driven by sensory-discriminative aspects such as perceived stimulus intensity, cognitive aspects such as arousal or attention, or emotional aspects such as intensity of unpleasantness. The specific balance of processes contributing to the overall BOLD response is uncertain. Decomposing measured neural responses to noxious stimuli into constituent components is an important but challenging task. Control stimuli that are matched to the noxious stimulus in all non-nociceptive aspects could help to isolate nociceptive contributions to the signal. However, these are

difficult to achieve in adults, and even more so in infants, and it is arguably more pressing from a clinical perspective to understand the neurophysiological basis for the overall multifaceted response than an individual constituent component of the signal.

Several infant clinical pain scales are multidimensional in nature[2], such as the PIPP-R scale which takes into account behavioural and autonomic responses. Previous results from our lab demonstrate that pinprick-evoked brain responses measured with EEG (electroencephalography) correlate with both reflex behaviour assessed with EMG (electromyography)[3] and autonomic heart rate responses[7]. However, technical and safety challenges in incorporating EMG into the MRI environment and unreliable estimates of inter-subject variability in autonomic responses (due to the small effect size associated with the pinprick stimulus), precluded linking the BOLD response amplitudes to these behavioural and autonomic dimensions at the individual level in the current study. But it is important to note that these behavioural and autonomic responses lack specificity to pain dimensions[2,40], and thus cannot be used to decompose the noxious-evoked BOLD response into sensory-discriminative, cognitive, or emotional contributions, or demonstrate that evoked BOLD responses differentially reflect nociception.

Here, we adopted a template-based approach to demonstrate that variability in noxious-evoked activity in neonates shows concordance with adult fMRI activity associated with pain and nociception, and demonstrates the expression of the NPS, which

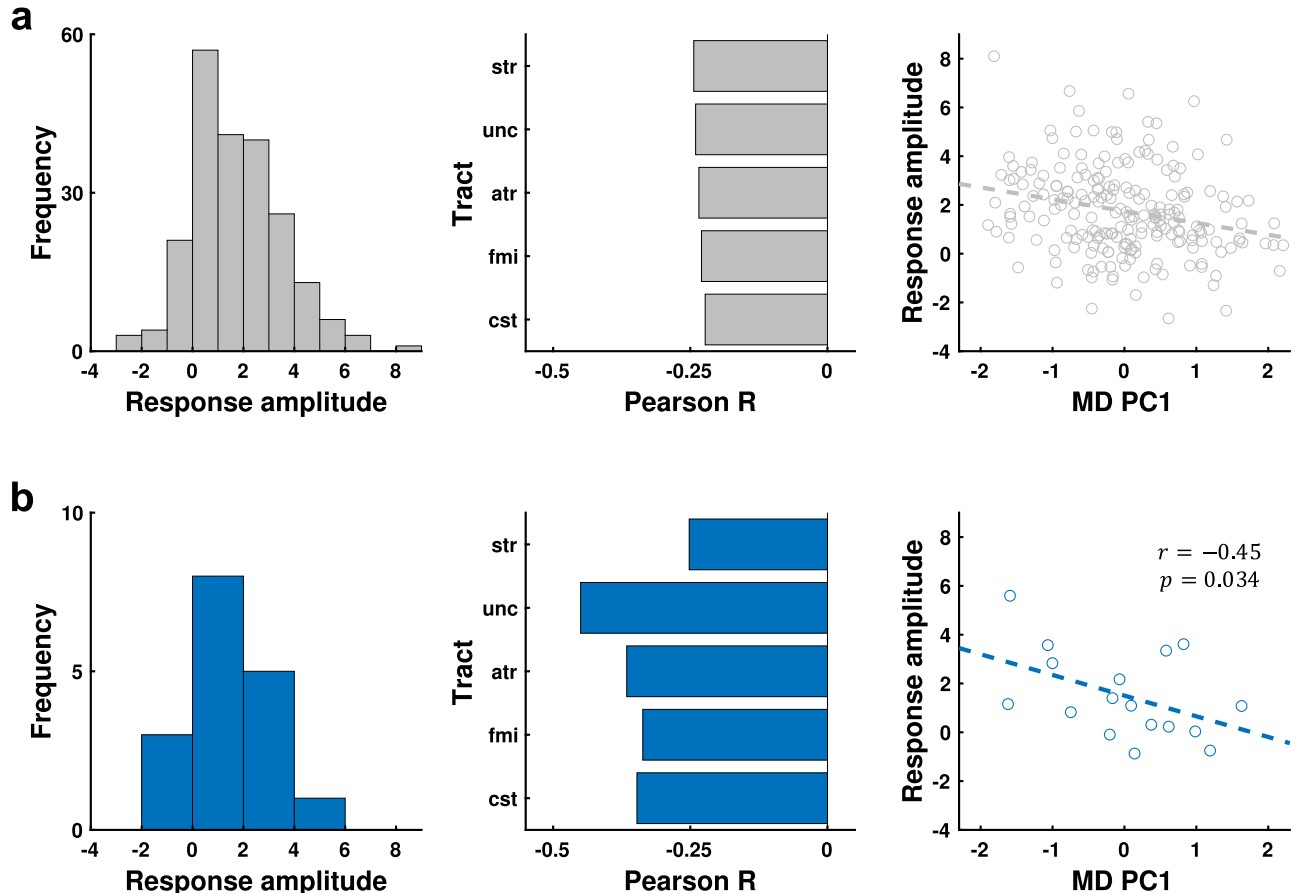

**Fig. 6 Confirmation of negative associations between noxious-evoked response amplitudes and white matter mean diffusivities in the noxious stimulation paradigm dataset. a** Results using predicted responses in dHCP dataset (*n* = 215). **b** Results using observed responses in noxious stimulation paradigm dataset (*n* = 17). **a**, **b** Left: histograms display the frequency distributions of the noxious-evoked response amplitudes. Middle: bar plots displaying the Pearson correlation coefficients between noxious-evoked response amplitudes and MD for the five white matter tracts identified in the exploratory arm analysis (Fig. 5). Right: scatter plots displaying the negative correlation between noxious-evoked response amplitudes (*y*-axis) and MD PC 1 (*x*-axis). Due to the negative correlation observed in the exploratory test in the dHCP dataset (**a**), the confirmatory Pearson correlation test in the noxious stimulation paradigm dataset (**b**) was one-sided with hypothesised negative correlation. These cross-dataset consistencies confirm the exploratory arm findings and establish initial validation for the underlying resting-state prediction model. atr anterior thalamic radiation, cst corticospinal tract, fmi forceps minor, str superior thalamic radiation, unc uncinate fasciculus, MD PC1 mean diffusivity principal component 1, *r* Pearson correlation coefficient, *p* *p*-value associated with *r*. Source data are provided as a Source Data file.

tracks adult verbal pain ratings[17]. These results provide important supporting evidence for the processes contributing to the noxious-evoked response, and gives us some confidence that the brain responses reflect a set of pain-related processes with some concordance with existing infant and adult pain measures.

Our cross-modality analyses revealed both local and global resting-state activity and white matter microstructural associations with noxious-evoked response amplitudes, such that neonate's with larger response amplitudes generally had larger RSN activity amplitudes, greater white matter FA, and lower white matter MD. RSN amplitudes exhibited a near-global positive correlation with noxious-evoked response amplitudes, with neonates' resting-state SMN activity being most robustly coupled to noxious-evoked response amplitude. White matter MD and FA exhibited a near-global negative and positive correlation with noxious-evoked response amplitudes, respectively, suggesting a greater degree of myelination or fibre packing density could underpin larger responses[41,42]. The most robust structure-function associations were negative correlations between response amplitudes and the MD of five functionally relevant tracts: superior and anterior thalamic radiations, corticospinal tract, uncinate fasciculus, and forceps minor.

This subset of tracts was intimately functionally related to both our noxious stimulation paradigm and the SMN. The superior thalamic radiations relay incoming nociceptive input from the somatosensory thalamus to primary somatomotor cortical regions, and corticospinal tracts output motor signals, such as nocifensive actions. Together, these tracts form important structural connections for somatomotor aspects of pain processing[43]. The anterior thalamic radiations and uncinate fasciculi connect limbic system regions involved in emotional and cognitive aspects of pain and nociceptive processing[43]. Finally, the forceps minor connects medial and lateral frontal lobes of left and right hemispheres, brain regions involved in higher-order integration and modulation of different aspects of pain signalling[15]. Due to practical and ethical experimental design limitations inherent to researching neonatal pain, the multimodal MRI design adopted in this study was an invaluable approach to advancing our understanding of structural and functional architectural features relevant to neural processing of noxious stimuli.

The above set of global and local functional and microstructural correlates of noxious-evoked response amplitudes, taken together, suggest neonates' noxious-evoked response amplitudes may reflect brain maturity, with larger response

amplitudes indicating a more mature brain. A number of published studies support this interpretation.

Increasing tactile-evoked BOLD responses is a general developmental trend from infancy to adulthood[44]. Similarly, in studies using NIRS (near-infrared spectroscopy) to measure haemodynamic responses to pain in the perinatal period, amplitudes of these responses progressively increased with age[13]. In this maturity framework, our observed negligible or negative BOLD responses could be explained by immaturity in neurovascular coupling mechanisms, such as vasoconstriction or insufficient functional hyperaemia to meet oxygen demands[45]. However, there are several mechanisms by which negative BOLD effects can be seen in adults[46]. As the three negative BOLD responses we observed were generally of low magnitude, they may simply reflect the effects of acquisition noise on a limited BOLD response. As observed in fMRI-based resting-state studies, functional connectivity strength and activity amplitude in sensory, motor, and cognitive network all increase with age[47,48], with spatial and functional heterogeneity in timing and rate of development observed for both cortical structural and functional maturity[42,49]. Similarly observed in white matter microstructure studies, white matter decreases in MD and increases in FA are global developmental trends observed during the perinatal period[35] and from infancy through to adulthood[41]. Analogous to cortical development, white matter maturation is both spatially and functionally heterogeneous in timing and rate of development[41,42,50]. These global developmental increases in RSN amplitudes, increases in white matter FA, and decreases in white matter MD coupled with regional and functional variations could then explain our findings of global resting-state activity and white matter microstructure associations coupled with most robust associations identified in functionally relevant networks and tracts.

In addition, this neural maturity interpretation is consistent with the suggestion that noxious-evoked responses reflect a relatively stable trait feature. Neural maturity may then be a common cause between neonates' resting-state and noxious-evoked activities mediating our resting-state based predictions of noxious-evoked response amplitudes.

Age is a good but imperfect proxy for neural maturational state assessment due to non-linearities in associations between age and maturation, an extreme pathological example being cases of developmental dysmaturity[51]. In this study, it was not the youngest born or imaged neonates, or those with the lowest BW, who had the smallest responses, suggesting individual variability in brain maturity is not fully captured by indicators such as age and brain volume (Supplementary Fig. 4). It is likely these features would increase their predictive value when considering a wider age range, so we do not conclude that age and brain volume are irrelevant to understanding neonates' noxious-evoked response amplitudes. How well this maturity interpretation generalises to neonates outside the studied age-range or to non-normative populations, such as those born very prematurely, would be an informative route of enquiry.

The findings of this study may have valuable ramifications in both research and clinical contexts. We applied our prediction model to a large independent age-matched dHCP sample, generating predicted noxious-evoked response amplitudes per neonate from their resting-state activity. The distribution of predicted response amplitudes in the dHCP dataset closely matched the measured response amplitudes of our noxious stimulation paradigm dataset, and several of the global and local structure-function association patterns initially identified in the dHCP dataset were subsequently confirmed in our noxious stimulation paradigm dataset. These cross-dataset consistencies existed despite differences in scanner hardware and acquisition protocols,

confirming that our prediction model generalised well to the dHCP data, establishing initial indirect external model validation. It thus appears to be possible to generate biologically meaningful noxious-evoked response amplitudes in a large publicly available dataset lacking noxious-evoked BOLD response data, but which features multimodal MRI data and a rich catalogue of interesting demographic variables. This could provide a platform for future MRI-based neuroscientific research, which would significantly advance our ability to study and understand neonatal pain system development.

We demonstrate that it is possible to predict neonates' cerebral noxious-evoked responses from resting-state activity, and that the noxious-evoked responses reflect a relatively temporally stable trait effect. Both of these features are potentially desirable from a clinical perspective. Due to the brief extra-uterine medical history of neonates, predicting pain outcomes from monitoring spontaneous brain activity may be able to provide valuable personalised insight that could reduce the degree of uncertainty in clinical decision-making regarding the treatment of neonatal pain. Predictions with relative temporal stability (e.g. over a 24 h window) will likely be valuable in developing predictive approaches with clinical utility to ensure insensitivity to transient state effects such as wakefulness. However, noxious-stimulation fMRI is a research tool lacking meaningful clinical utility for infant pain assessment. Thus, translation of brain-function-based prediction models to common cot-side imaging techniques such as haemodynamic-based NIRS or neurodynamic-based EEG could significantly reduce the gap between MRI research outputs and clinical utility. If the findings of the current study are translatable to cot-side imaging modalities, this could significantly progress the development of brain-based predictive approaches to tailoring pre-procedural or pre-operative decision-making regarding neonatal pain management strategies.

Our reported functional coupling between resting-state and stimulus-response activities is currently limited to the noxious stimulus employed in this study. However, in adults this functional coupling has been demonstrated for a wide range of tasks[26,29,52], and we imagine a similar generalisability of resting-state coupling to stimulus responses may be applicable to neonates. While "task" fMRI experimental designs are limited in neonates, previous studies using non-nociceptive stimuli, such as non-noxious touch[53], auditory[54], and visual[55] stimuli have demonstrated the feasibility of multimodal experimental designs to test this directly. In addition, the functional coupling results may not generalise to premature neonates younger than 35.9 weeks PMA, the youngest subject included in the present cohort, as these younger neonates will have poorer neurovascular coupling[45,56], which would need to be taken into account.

Our RSN amplitudes used for prediction were extracted from nine large-scale networks. While these networks cover multiple sensory modalities and cognitive networks, they do not provide a whole-brain parcellation. Increasing brain coverage to include pain-relevant regions such as the amygdala and orbitofrontal cortex and networks such as the salience network would likely improve the accuracy and generalisability of neonatal resting-state based-predictions. Furthermore, as part of a recent study in adults demonstrating the predictability of individual pain sensitivity from resting-state functional connectivity[31], the researchers have developed a Resting-state Pain susceptibility Network (RPN) signature that may also have predictive value when applied to neonates, analogous to the application of the adult NPS[37] to the neonatal population in the current study and a recent publication[17].

Finally, we identified structure-function associations related to noxious stimulus processing using representational dMRI models (DTI/DKI). While the combination of MD, FA, and MK contain

all signal features available to the biophysical model NODDI[57,58], future work applying both representational and biophysical models in complementary fashion would likely aid biophysical interpretability of the microstructural phenomena underpinning our observed structure-function associations.

Even healthy neonates within the first few days of postnatal life display a wide range of responses to noxious input, likely a result of both genetic and environmental influences[59]. This normative variability may reflect differences in individual resilience and vulnerability to environmental insults, such as clinical painful procedures that are frequently performed in hospitalised neonates[1]. The ability to predict a neonate's pain responses may have the potential to advance personalised pre-emptive pain management, and this study highlights the importance of understanding the underlying brain microstructure and resting-state neurophysiology in achieving this goal. A better under-standing of how individual differences in brain architecture influence pain processing is of paramount importance if we are to identify neonates at increased risk of long-term alterations in brain structure and function and cognitive performance as a result of early life pain exposure. Early life pain and stress have the potential to alter a neonate's developmental trajectory and to influence their well-being in childhood[60,61], but it may also increase the risk of developing chronic diseases in later life[62,63]. The development of brain-based pain predictors could help identify these vulnerable neonates and guide the tailoring of pain relief treatments in a more principled, personalised, and evidence-based manner.

## Methods

**Subject information**. We recruited healthy neonates from the John Radcliffe Hospital postnatal ward (Oxford University Hospital NHS Trust). Neonates were considered healthy if they were inpatients that never required admission to the neonatal unit, had no history of congenital conditions or neurological problems, and were clinically stable at the time of the study. Written informed consent was obtained from parents prior to the study. Ethical approval was obtained from an NHS Research Ethics Committee (National Research Ethics Service, REC reference: 12/SC/0447), and research was conducted in accordance with Good Clinical Practice guidelines and the Declaration of Helsinki.

Twenty-one neonates had noxious-evoked and resting-state data collected. Subjects were excluded from analysis if scan runs were not fully completed (to remove inter-subject variability in data quality related to scan length) or the vertex of the cerebral cortex left the scan field of view more than 5% of the scan run (to ensure reliable data in this functionally relevant brain region). Using these criteria, three subjects were excluded resulting in a sample of $n = 18$ neonates. Sample demographic details are described below (see "Clinical variables").

**Experimental design**. Scans occurred in the Wellcome Centre for Integrative Neuroimaging (Oxford, UK). Neonates were fed and swaddled, fitted with ear protection, then placed on a vacuum-positioning mattress with padding around the head. Heart rate and blood oxygen saturation were monitored throughout scanning, but were not of sufficient quality for analysis. The noxious stimulation paradigm was an event-related design in which a mild non-skin-breaking noxious stimulus, the 128 mN sharp-touch pinprick (PinPrick Stimulator, MRC Systems), was applied to the dorsum of the left foot: ten trials, 1 s per trial, 25 s minimum inter-stimulus interval[10]. Stimulus time-locking was performed using Neurobehavioural Systems software v20.1 (https://www.neurobs.com/). Stimuli were applied when neonates were naturally still to minimise the influence of motion artefacts. For all other scan types, neonates lay passively in the scanner. No sedatives were used at any stage.

**MRI data acquisition**. All data were collected on a 3T Siemens Prisma with an adult 32-channel receive coil using the following scan parameters. Structural: T2-weighted, TSE (factor 11), 150° flip angle, TE = 89 ms, TR = 14,740 ms, GRAPPA 3, 192 × 192 in-plane matrix size, 126 slices, 1 mm isotropic voxels, acquisition time (TA) = 2 mins 13 s. Fieldmap: gradient echo, 2DFT readout, dual-echo TE1/TE2 = 4.92/7.38 ms, TR = 550 ms, 46° flip angle, 90 × 90 in-plane matrix size, 56 slices, 2 mm isotropic voxels, TA = 1 min 40 s. Resting-state and noxious stimulation fMRI: T2* BOLD-weighted, gradient echo, EPI readout, 70° flip angle, TE = 50 ms[64], TR = 1300 ms, multiband 4[65,66], 90 × 90 in-plane matrix size, 56 slices, 2 mm isotropic voxels, AP phase encode direction. Resting-state TA = 10 min 50 s (500 volumes), mean noxious stimulation TA = 6 min approx. (277 volumes approx.). Diffusion MRI: T2 diffusion-weighted, spin echo, EPI readout, 90° flip angle, TE = 73 ms,

TR = 2900 ms, multiband 3, 102 × 102 in-plane matrix size, 60 slices, 1.75 mm isotropic voxels, AP phase encode direction, multishell ($b$ = 500, 1000, 2000 s/mm²), 143 directions uniformly distributed over the whole sphere, TA = 8 min. Phase-reversed b0 images were collected to derive a spin-echo fieldmap for distortion correction of diffusion data. The entire MRI acquisition protocol had a nominal duration of 40 min.

**MRI data preprocessing**. All MRI data were preprocessed using dHCP pipelines. The T2 structural data were processed using the MIRTK Draw-EM neonatal pipeline v1.1[67,68]. Noxious stimulation and resting-state fMRI were pre-processed using the dHCP fMRI pipeline v0.5.3[47,69]. Data were motion and distortion corrected using FSL's EDDY[70,71], which included slice-to-volume motion correction[72] and susceptibility-by-movement distortion correction[73]. Noxious stimulation data were high-pass temporally filtered at 0.01 Hz, and resting-state data at 0.005 Hz. Data were denoised using FSL's FIX[74,75], low-pass spatially filtered with a 3 mm FWHM filter using FSL's SUSAN[76], and scaled to a common global spatiotemporal median. For spatial normalisation, data were registered from functional to structural space using BBR[77] with FSL's FLIRT[78,79], then from structural space to the 40-week standard template[80] using ANTs's SyN[81].

Diffusion data were analysed using the dHCP dMRI pipeline v0.0.2[35,82]. Phase-reversed field maps were processed using FSL's TOPUP[83,84]. Data were corrected for motion, distortion, and eddy currents using FSL's EDDY, which included outlier detection and replacement[85], slice-to-volume motion correction, and susceptibility-by-movement distortion correction. Spatial normalisation followed the same sequence of registrations as the functional data.

**Noxious-evoked response amplitudes**. Noxious-evoked response maps were generated using subject-level voxelwise GLM analysis in FSL's FEAT[86], fitting the term-neonate double-gamma HRF[44,69]. A group average t-statistic map was generated using the 18 subjects' regression parameter maps. Regression parameter maps were used as subject-level response maps; the group-average t-statistic map was used as the group-level response map. The group-level response map was regressed onto each subject-level response map producing a spatial regression coefficient, constituting each subject's overall noxious-evoked response amplitude (Supplementary Fig. 10 Step 1). The influence of HRF goodness-of-fit on noxious-evoked response amplitudes was assessed in Supplementary Information (Supplementary Figs. 1 and 3).

To localise the noxious-evoked activity, we generated a thresholded group average activity map using group-level voxelwise GLM analysis in FSL's Randomise[87]. Statistical significance was assessed using permutation testing with 10,000 permutations, variance smoothing (6 mm FWHM kernel) due to limited degrees of freedom[88], cluster-based thresholding with $z = 3.1$ ($p = 0.001$) cluster-defining threshold[89], and a FWER-corrected cluster p-value of $p = 0.05$. Regions of activity were identified using the probabilistic Harvard-Oxford Cortical Structural Atlas[90].

To gain insight into the processes underlying the noxious-evoked activity, we assessed the expression of several adult pain signature templates and Neurosynth meta-analysis association test map templates by performing whole-brain correlations between adult template maps and subject-level response maps (Supplementary Fig. 10 Step 2). For each template, $n = 18$ correlation coefficients were generated, and group average template expression (average correlation) was assessed using two-tailed t-tests with statistical significance assessed non-parametrically in FSL's PALM[87] using 10,000 permutations.

The adult pain signatures tested were the NPS and Social Rejection Pain signature, where the social rejection template was used as a negative control, as per the original NPS adult study[37]. Association test maps were used for all Neurosynth terms, as these maps display brain regions that are preferentially related to the term-of-interest. The primary Neurosynth term-of-interest was "pain"; "visual" was used as a Neurosynth negative control; a subset of pertinent pain dimensions was assessed, including the sensory-discriminative term "nociceptive", cognitive terms "arousal", "salience", and "attention", and the emotional term "unpleasant". Due to the whole-brain nature of the expression correlations and the default thresholded nature of all adult template maps included, we used the thresholded group average activity map generated from the neonates' noxious-evoked responses (thresholding performed in Randomise as described above) as the positive control. While the correlation between this thresholded noxious-evoked response map and the neonates' response maps is likely inflated due to circular analysis, the correlation strength of this positive control sets a useful upper limit reference.

Finally, inter-subject variability in noxious-evoked response amplitudes (regression parameters) and adult template correspondences (correlation coefficients) were assessed for all functional templates using two-tailed Pearson correlation tests with statistical significance assessed using 10,000 permutations.

**Noxious stimulation imaging confounds**. In all analyses using noxious-evoked responses, amplitudes were adjusted for mean head motion (mean framewise displacement), stimulus-correlated head motion (multiple correlation coefficient between the predicted BOLD response, i.e., stimulus timeseries convolved with the HRF, and the 24 head motion timeseries estimated during preprocessing), and CSF signal amplitude (mean regression coefficient within the CSF ROI, intended to

capture residual cardiac pulsatility). Details on ROI construction are provided in Supplementary Information (Supplementary Fig. 8).

**RSN amplitudes**. To define a robust set of RSNs, RSNs identified in the noxious stimulation paradigm dataset ($n = 18$) were compared to those identified in a dHCP dataset previously produced as part of the dHCP[38] (Supplementary Fig. 10 Step 3). Robust networks were defined as those replicated across datasets to ensure networks were not dataset-specific. This dHCP dataset included 242 healthy term-aged neonates: mean GA at birth = 38.6 weeks; mean PMA at scan = 40.4 weeks; 112 females[38]. The RSN analysis performed on our noxious stimulation paradigm dataset was matched to that of the dHCP dataset[38,47]. In brief, PFM analysis using FSL's PROFUMO[91,92] was run with pre-specified dimensionality of 25, using the term-neonate double-gamma HRF[44,69] as the temporal prior. PROFUMO's Bayesian model complexity penalties eliminated modes unsupported by the data, thus returning a number of group-level modes less than the pre-specified dimensionality. The data-determined dimensionality for the noxious stimulation paradigm dataset was 11, nine of which corresponded to the dHCP dataset RSNs assessed using spatial correlation followed by visual confirmation. Due to the larger sample size, the dHCP RSN maps had greater SNR and were thus used throughout as the nine RSN template maps.

These RSN template maps were spatially regressed onto each neonate's resting-state data using multiple regression, resulting in network timeseries. Timeseries amplitudes were quantified as the MAD, due to MAD's increased robustness to outliers compared to the more commonly used standard deviation (Supplementary Fig. 10 Step 4). The association between network timeseries outliers and noxious-evoked response amplitudes was assessed in Supplementary Information (Supplementary Figs. 2–3).

**Resting-state imaging confounds**. In all analyses using RSN amplitudes, amplitudes were adjusted for head motion (mean framewise displacement), CSF amplitude (timeseries MAD extracted using the CSF ROI, intended to capture residual cardiac pulsatility) and white matter amplitude (timeseries MAD extracted using the white matter ROI, intended to capture global signal amplitude). Details on ROI construction are provided in Supplementary Information (Supplementary Fig. 8). These confounds were also directly tested for association with noxious-evoked response amplitudes.

**Clinical variables**. Six clinical variables were tested for association with noxious-evoked response amplitudes: PMA, GA, PNA, BW, TBV, and sex. Age variables are defined according to the American Academy of Paediatrics[93], and TBV was calculated from neonates' structural MRI tissue segmentation outputs. This group ($n = 18$) included 10 males, and means (and standard deviations) for numeric variables were: 38.3 (1.8) weeks GA, 38.7 (1.7) weeks PMA, 2.8 (2.3) days PNA, 3.34 (0.69) kg BW, and 289,823 (49,021) mm$^3$ TBV. Testing the clinical variables assessed whether associations between RSN amplitudes and noxious-evoked response amplitudes could be explained by these biologically interesting variables.

**Predicting noxious-evoked response amplitudes**. For prediction analyses (Supplementary Fig. 10 Step 5), the primary responses to be predicted were the neonates' overall noxious-evoked response amplitudes (Fig. 1 scalar values). Three sets of predictors were tested: nine RSN amplitudes, six clinical variables, and three resting-state imaging confounds. To assess whether pain components of the overall noxious-evoked response amplitudes can be predicted from resting-state data, we assessed RSN amplitude-based predictions for both the NPS and Neurosynth pain map response magnitudes. Adult pain signature response magnitudes were

quantified using cosine similarity, which is equivalent to the Pearson correlation coefficient without mean-centring, thus retaining magnitude information.

For each set of predictors, a multivariate linear SVR model was used to generate predictions using leave-one-out cross-validation (LOO-CV). The linear SVR model was fit using scikit-learn v0.21.3 packages[94] (Python v3.7.4). Noxious-evoked, NPS, and Neurosynth pain responses were adjusted for the noxious-evoked response imaging confounds, and RSN amplitudes were adjusted for the resting-state imaging confounds using cross-validated confound regression (https://github.com/lukassnoek/MVCA)[95]. SVR parameters were: kernel=linear, loss=epsilon insensitive, epsilon = 0.1, regularization=ridge, regularization strength = {0.001, 0.01, 0.1, 1}. Regularization strength optimisation was performed using an initial LOO-CV grid search.

Prediction performance was assessed using sums-of-squares coefficient of determination ($R^2$) as primary outcome of interest, as well as root mean squared error (RMSE) and Spearman's rank correlation coefficient ($R_{Sp}$). We performed one-tailed significance tests for these measures using permutation tests, running 10,000 permutations through the full prediction pipeline. The relationship between individual predictors and noxious-evoked response amplitudes was assessed in Supplementary Information (Supplementary Fig. 4).

**Structure-function analysis using an exploratory-confirmatory approach**. The neonates' noxious-evoked response amplitudes were assessed for structure-function associations by analysing white matter microstructure. Due to the lack of research into the microstructural basis of neonates' noxious-evoked responses, an exploratory analysis was required. However, due to the small sample size of the noxious stimulation paradigm dataset ($n = 17$, one subject was excluded from structure-function association analyses due to incomplete dMRI data), appropriate corrections for multiple testing would prohibit the identification of true positives. We thus adopted a two-armed two-dataset exploratory-confirmatory analysis approach. In the initial exploratory arm, we used a large age-matched dHCP sample to test a range of microstructural features to identify candidate white matter tracts and diffusion model parameters (Supplementary Fig. 10 Step 6). Structure-function relationships identified here were used to formulate data-driven hypotheses. In the confirmatory arm, these hypotheses were subsequently tested for validation in the noxious stimulation paradigm dataset (Supplementary Fig. 10 Step 7). Cross-dataset consistencies in structure-function associations constitute initial indirect external validation for the resting-state prediction model, as these consistencies rely on the predicted response amplitudes in the dHCP dataset being similar in nature to the true response amplitudes in the noxious stimulation paradigm dataset.

**dHCP dataset sample selection**. Neonates were included in our dHCP sample[96] if data were of reasonable quality and neonates were age-matched to our noxious stimulation paradigm dataset. The three data quality criteria were: both fMRI and dMRI data passed dHCP QC pipelines[47,82], both scan sessions completed fully, and the vertex of the cerebral cortex remained within the scan field of view for at least 95% of the scan session. The two age criteria were: neonates were between 36 and 42 weeks for both GA and PMA, and neonates were scanned within the first 10 postnatal days. These criteria resulted in a sample of $n = 215$ neonates: 122 males, mean (and standard deviations) age at scan was 40.1 (1.4) weeks PMA. Key functional and diffusion acquisition parameters for this dHCP dataset are displayed in Table 3 to facilitate comparison with our noxious stimulation paradigm dataset.

**dHCP dataset noxious-evoked response amplitudes**. The dHCP dataset does not include noxious stimulation paradigm data. To analyse structure-function relationships relevant to noxious-evoked responses, the dHCP resting-state data

**Table 3 Comparison of key data acquisition parameters between the noxious stimulation paradigm (local) dataset and dHCP dataset for resting-state fMRI and dMRI data.**

|  | Parameter | Local dataset | dHCP dataset |
|---|---|---|---|
| Scanner | Model | Siemens Prisma | Philips Achieva |
|  | Field strength | 3 T | 3 T |
| Head coil | Channel density | 32-channel | 32-channel |
|  | Size | adult | neonatal |
| Resting-state | Echo time, TE (ms) | 50 | 38 |
|  | Repetition time, TR (ms) | 1300 | 392 |
|  | Multiband acceleration factor, MB | 4 | 9 |
|  | Voxel size, isotropic (mm) | 2 | 2.15 |
|  | Number of volumes | 500 | 2300 |
| Diffusion | Echo time, TE (ms) | 73 | 90 |
|  | Repetition time, TR (ms) | 2900 | 3800 |
|  | Multiband acceleration factor | 3 | 4 |
|  | Voxel size, isotropic (mm) | 1.75 | 1.5 |
|  | Number of volumes | 163 | 300 |
|  | b-values (s/mm$^2$) | 0, 500, 1000, 2000 | 0, 400, 1000, 2600 |
|  | Number of volumes/directions | 20, 23, 50, 70 | 20, 64, 88, 128 |
|  | Gradient duration, δ (ms) | 12.5 | 14 |
|  | Gradient separation, Δ (ms) | 35.5 | 42.5 |

were mapped to noxious-evoked response amplitudes using the RSN amplitude-based prediction model described above. This model was trained on the noxious stimulation paradigm dataset ($n = 18$) using the nine confound-adjusted RSN amplitudes as predictors and the confound-adjusted noxious-evoked response amplitudes as responses. In our dHCP dataset ($n = 215$), predictors and confounds were extracted in an identical manner to the noxious stimulation paradigm dataset analysis, and the RSN amplitudes were used to generate predicted noxious-evoked response amplitudes.

**White matter microstructural features**. Our dHCP sample ($n = 215$) was used to generate 16 bilateral white matter tracts using the "baby autoPtx" approach established as part of the dHCP dMRI preprocessing pipeline development[35]. In brief, FSL's probabilistic multi-shell ball and zeppelins model[97] is fit, then probabilistic tractography using FSL's PROBTRACKX[98,99] is run using pre-defined seed, target, and exclusion masks. At the time of analysis, 29 tracts were available, 13 unilateral and three bilateral. Unilateral tracts were fused to create bilateral tracts analogous to our bilateral RSNs, resulting in a total of 16 bilateral tracts: acoustic radiation, anterior thalamic radiation, cingulate gyrus part of the cingulum, parahippocampal part of the cingulum, corticospinal tract, forceps major, forceps minor, fornix, inferior fronto-occipital fasciculus, inferior longitudinal fasciculus, middle cerebellar peduncle, medial lemniscus, posterior thalamic radiation, superior longitudinal fasciculus, superior thalamic radiation, uncinate fasciculus. Normalised probability value results of each tract were group-averaged in standard space and thresholded at a probability of 0.01.

We used FSL's DTIFIT to generate MD, FA, and MK parameter maps. We used the MD and FA maps from the b1000 shell, and MK maps from all three shells in both datasets. We thresholded each parameter map to remove noisy voxels with values outside expected theoretical ranges, likely due to poor SNR or head motion: negative values for MD; values outside [0,1] for FA; values outside [0,3] for MK. Mean parameter values for each tract constituted the white matter microstructural features for our structure-function analyses. Comparisons of these microstructural features between the noxious stimulation paradigm and dHCP datasets is assessed in Supplementary Information (Supplementary Fig. 5).

**Identifying nociception-related structure-function associations**. Using the dHCP sample ($n = 215$) for the initial exploratory arm analyses, univariate correlations between predicted noxious-evoked response amplitudes and each microstructural feature was assessed using two-tailed Pearson correlations adjusted for three dMRI imaging confounds: mean head motion, number of noisy voxels outside expected theoretical ranges, and TBV. Adjustment for TBV mitigates macrostructural tissue density and partial volume effect confounds, which bias dMRI microstructural parameters. Statistical significance was assessed using 10,000 permutations and FWER-corrected for multiple testing across all 48 tests (16 tracts × 3 parameters)[100]. While significant correlations are statistically valid, they are tentative due to the use of predicted noxious-evoked response amplitudes without a known ground truth.

Due to the presence of consistent structure-function correlation polarities across tracts identified in the dHCP dataset, variance across multiple tracts for a single dMRI parameter was pooled using principal component analysis (PCA). The first principal component across these tracts (e.g. MD PC1) was assessed for associations with noxious-evoked response amplitudes using two-tailed Pearson correlations, significance assessed using 10,000 permutations. In the dHCP dataset, these effect size and statistical significance measures are biased due to circularity in explanatory variable selection[39], but the bias is restricted to the exploratory arm analyses.

In the subsequent confirmatory arm analyses, the two hypotheses formulated in the dHCP dataset (see Results) were tested in the noxious stimulation paradigm dataset. First, the polarity of the Pearson correlation coefficient between noxious-evoked response amplitudes and individual tracts were qualitatively assessed. While similar correlation polarities alone are not strong evidence, when considered in combination with other evidence, this correlation polarity information provides useful complementary insight. Second, the correlation between noxious-evoked response amplitudes and microstructural features of specific subsets of tracts was assessed using one-tailed Pearson correlation tests, the directionality of the tailed test hypothesised by the nature of the exploratory arm results. Statistical significance was assessed using 10,000 permutations. The effects of both global and local structure-function association effects observed in both the dHCP and noxious stimulation paradigm datasets were further assessed in Supplementary Information (Supplementary Figs. 6–7).

**Reporting summary**. Further information on research design is available in the Nature Research Reporting Summary linked to this article.

## Data availability
The noxious stimulation paradigm data that support the study findings are available from the corresponding author upon reasonable request. Due to ethical restrictions, it is appropriate to monitor access and usage of the data as it includes highly sensitive information. Data sharing requests should be directed to rebeccah.slater@paediatrics.ox. ac.uk. The dHCP data (Second Data Release) that support the study findings are available online (http://www.developingconnectome.org/second-data-release). Source data are provided with this paper.

## Code availability
Functional and diffusion data preprocessing pipelines are available via the dHCP (http://www.developingconnectome.org). Customisations of the dHCP fMRI pipeline used for our noxious-stimulation paradigm data are described here (https://doi.org/10.1016/j. neuroimage.2018.11.006), and were implemented using FEAT and other standard FSL tools, available online (https://fsl.fmrib.ox.ac.uk/fsl/fslwiki). Resting-state networks were identified using FSL's PROFUMO (https://git.fmrib.ox.ac.uk/samh/profumo), and DTI/DKI model fitting was performed using FSL's FDT (https://fsl.fmrib.ox.ac.uk/fsl/fslwiki/FDT). Tractography was performed using Baby AutoPtx as described here (https://doi.org/10.1016/j.neuroimage.2018.05.064). Cross-validated confound regression was performed as described here (https://doi.org/10.1016/j.neuroimage.2018.09.074) and is available online (https://github.com/lukassnoek/MVCA).

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

## Acknowledgements

This study was funded by a Senior Wellcome Research Fellowship awarded to R.S. (207457/Z/17/Z) and by a BLISS research grant, which funds L.B. This study was sup-
ported by the Wellcome Centre for Integrative Neuroimaging, which is supported by core funding from the Wellcome Trust (203139/Z/16/Z). The dHCP data were provided by the developing Human Connectome Project, KCL-Imperial-Oxford Consortium funded by the European Research Council under the European Union Seventh Framework Programme (FP/2007–2013)/ERC Grant Agreement no. [319456]. We are grateful to the families who generously supported this trial. We are grateful for the provision of simultaneous multi-slice (multi-band) pulse sequence and reconstruction algorithms from the Centre for Magnetic Resonance Research, University of Minnesota.

## Author contributions

L.B. conceived the idea for the study, analysed the data, interpreted the results, and wrote the paper. F.M. collected the data and revised the paper. S.F. provided technical assistance with the dHCP fMRI preprocessing pipeline, provided the dHCP resting-state network maps, and revised the paper. M.A. analysed the data and revised the paper. R.M. analysed the data and revised the paper. M.B. provided technical assistance with the dHCP dMRI preprocessing pipeline, provided the dHCP white matter tract maps, and revised the paper. R.R. interpreted the results and revised the paper. S.J. designed the experiment, interpreted the results, and revised the paper. E.D. designed the experiment, interpreted the results, and revised the paper. R.S. conceived the idea for the study, designed the experiment, interpreted the results, and wrote the paper.

## Competing interests

The authors declare no competing interests.
