## [Peer Review File · Nature Communications]

Reviewer #1 (Remarks to the Author):

The authors purport to reveal brain mechanisms for noxious stimuli in infants. A fixed force pin prick stimulus related brain activity is studied in 18 infants. Responses are then related to resting state network properties within and across independent data sets. Moreover, the responses are related to white matter myelination properties.

The data analysis seems rigorous and the observed activity seems genuinely a BOLD response. The authors extensively and comprehensively address worries that the collected signal could be artifactual.

Similarly relating these responses to resting state and to DTI parameters are properly executed.

However, the paper suffers from over-interpretation and this is a major flaw.

Given the study has no control stimulus to which observed brain activity can be contrasted with, we really have no idea what the observed brain activity responses correspond to. Moreover, it is clear that the activity is spread throughout the brain, and in 3 subjects responses are negative. Yet, the authors interpret all of this as nociceptive and pain related brain activity. Far simpler interpretation would be that these are simply arousal signals. The infant is exhibiting a generalized excitation. Or in the case of the 3 where negative responses are seen, they are showing diminished arousal. All the data the authors present are consistent and interpreted much more parsimoniously within such a framing. Otherwise the paper remains pure speculation.

As it stands, the manuscript's title, abstract, introduction and most of the discussion need complete reorganization.

Reviewer #2 (Remarks to the Author):

The main objective of this study is to better understand the inter-individual variability in pain perceptions in infants. The authors first aimed to predict functional MRI (fMRI) response maps to noxious stimulations from the amplitude of resting-state functional networks. Second, they aimed to relate noxious responses to the individual white matter microstructure. This multimodal MRI study, combining task-based and resting-state fMRI and diffusion MRI, was performed in 18 newborns, and it also relied on analyses performed on data of more than 200 infants from the developing Human Connectome Project (dHCP).

The question raised by this study is highly interesting and still very little explored in newborns. The data and analyses carried out allow us to test the hypotheses put forward. The methodologies used are precise, adequate and quite detailed. In addition, the article is really enjoyable to read. Nevertheless, there are a few points that need to be clarified as detailed below.

* My main concern is the way the "structure-function" relationships have been explored. Indeed, the prediction results based on resting-state networks are easy to understand and interpret since they put forward the somatomotor network (the interpretation is less obvious for the visual network). But the results for the diffusion MRI part are not completely clear to me. How could the 5 observed tracts be related to the pain system? What do the results mean compared with the reported resting-state networks? Furthermore, the analyses performed to address this question don't seem to me the most appropriate. Indeed, Supplementary Figure 7 (left) shows that the correlation coefficients are gradually decreasing, and considering only 5 out of 16 tracts is only a matter of statistical threshold considered (and correction considered for multiple comparisons?). Why not consider all the tracts in a prediction model, such as the one used with resting-state networks?

Moreover, the relationships might be more a question of the overall microstructure of the white matter rather than the microstructure of specific tracts. The measurements for the individual tracts are probably highly correlated with the overall level of maturation of the white matter. Is it this level that is most relevant for predicting noxious responses? Or is it the deviations to this level for specific tracts? Why not then consider the first component of PCA on all tracts for correlations with noxious responses? The fact that the "structure-function" correlations between mean diffusivity values and responses are variable (i.e. with different correlation coefficients) between the two

groups of newborns (Sup Fig 7) might support this hypothesis of low specificity of the tracts found.

* Second, one point not mentioned in the article concerns the differences in the acquisition protocols (for resting-state fMRI and diffusion MRI) between the group of neonates with activation fMRI data and the dHCP cohort. How can this influence measurements and results?

The authors show a good correlation between the resting-state networks identified in the 2 groups. Is it the same for the diffusion measurements of the tracts? For instance, the authors might evaluate whether the differences in diffusion parameters between tracts are similar across the 2 groups.

More minor comments are detailed below.

* The number of infants considered in the dHCP cohort needs to be clarified: is it 215 or 242? Are they all full-term newborns? Are their ages at birth and at MRI comparable to the 18 newborns of the study? In fact, resting-state networks and white matter microstructure differ between premature and full-term newborns.

* The term "infant" should be replaced by "newborn" when referring to babies of less than 28 days post-natal age.

* When the authors describe the predictive results for clinical factors, they might emphasize that consequently it is not the youngest born or imaged newborns, or those with the lowest birth weight, who have the most immature responses.

* The authors might show the results for other diffusion parameters (anisotropy, mean kurtosis) in supplementary information.

* Why not consider parameters from NODDI model in addition to DTI and DKI?

* Abstract

- The authors might precise which resting-state networks are the most important in the prediction, as well as which white matter tracts show correlations with noxious responses.

* Introduction

- I83-92: The order of sentences could be reviewed to improve comprehension.

- I93-94: This result is not that obvious.

It might be important to report which resting-state networks and which white matter tracts are used in the following analyses.

* Results

- I131: It's six networks instead of three.

- I135: Is it 242 infants of the dHCP cohort for resting-state fMRI or 215?

- I176: Here it would be interesting to better detail which networks contribute most to the prediction.

- I176: Beyond univariate correlation analyses, would it be possible to identify which features (i.e. networks) contribute most to the support vector regression model?

- I202-205: Here it would be interesting to detail which tracts are related to predicted noxious responses. How are these tracts consistent with the resting-state networks found previously?

* Discussion

The discussion is interesting and well conducted. However, it could include a discussion of the specificity of the results found with regard to the correlations between structural and functional measures (see also previous points).

* Methods

- I378: Subject information: How many newborns had to be included to get interpretable functional MRI data (task-based and resting-state) for 18?

Are the fMRI data the same as those included in the study by Gokstan et al (2015)?

- I379: What is the number of directions for each b value? How many volumes for b=0 were acquired?

- I423: What are the differences in acquisition protocols (for resting-state fMRI and diffusion MRI) between the newborn group of interest and the dHCP cohort?

In particular, from which b values was the tensor model estimated for each group? Might this have an influence on the results?

Would the repetition time (TR) have an influence on the characterization of resting-state networks?

- I553: The authors should specify why they consider only 9 resting networks (this is said in the results section, but it would be useful to repeat it in the methods section).

- I610: It might be appropriate to better introduce the reasoning behind this exploratory-confirmatory analysis approach in the introduction of the article.

- I662: The authors should detail which tracts are of interest.

- I702: Reporting similar correlation polarities doesn't seem to be significant evidence of similarity.
- * Figures
 - Fig 1 and 2: The colour scale should be added.
 - Fig 3: Abbreviations could be detailed.
- * Supplementary Information
 - I117-118: I am not convinced by the observation that "a clear pattern emerges for both MD and FA". What is the correlation, for each of the two parameters, between the correlation coefficients measured for the 16 tracts in the 2 groups?
 - I126-127: I agree with the observation that "a global effect common to all white matter tracts to varying degrees". It would then seem to me relevant to consider other analyses, as mentioned at the beginning of the review.
 - Sup Fig 2: It would be more appropriate to put the time scale in seconds, as in the previous figure. What do these numbers of outliers represent in percentage to the total number of images acquired for each subject?
 - Sup Fig 4: Are p values corrected for multiple comparisons performed?
 - Sup Fig 7: For mean diffusivity in the newborns group of interest, higher correlation coefficients are observed for many other tracts than the ones considered (as outlined by the dHCP cohort). What could that mean?
The authors might present the tracts in the same order for the different diffusion parameters (same order as for MD) to facilitate the comparisons across parameters.
 - Sup Fig 9: It might be useful to clarify here the usefulness of these regions.

Reviewer #3 (Remarks to the Author):

In this manuscript Baxter and colleagues investigate neurophysiological underpinnings of individual variability in noxious-evoked pain activity using resting state and white matter diffusion MRI in 18 healthy infants. The authors show that an infants' nociception related brain activity is associated with pain-free resting state activity and white matter microstructure. Further, the authors cross-validate the existence of resting state networks and their association with white matter microstructure in an independent data set of 215 infants from the human connectome project.

This paper addresses a scientifically and clinically very relevant topic. Knowledge regarding developmental aspects in the neurobiology of nociception and pain is sparse, but urgently needed given that early life exposure to noxious procedures is known to impact an infants developmental trajectory including the susceptibility for chronic diseases. This is a challenging investigation both from an experimental and methodological point of view and the authors should be applauded for acquiring this unique data set and their attempt to overcome inherent limitations of the small sample size by replicating parts of their results in data from the human connectome project.

I have several comments that should be addressed

The authors should consider to replace the term pain by nociception at several places in the manuscript (e.g. "infant's pain-related brain activity" in the abstract), or make clear that the use nociception related activity as a proxy of an infant's painful experience.

Both in the introduction and the discussion the authors allude to the potential clinical implications of their work. These statements should be either down-toned or elaborated in more concrete terms: how will multimodal MRI inform clinical decision making at an individual level? I am not a paediatrician but I doubt that MRI will ever become a routine diagnostic in infants.

While I can fully follow the rationale to replicate the identification of infants' resting state and structural networks in an independent cohort, I have difficulty to assess the validity of the predicted amplitudes of noxious-evoked brain activity in the dataset from the human connectome project. Please comment. Also issues of circularity should be addressed if applicable.

The authors use head motion, CSF amplitude and white matter amplitude as potential confounders of the resting state analysis. Please also consider age and total brain volume as a covariate in the

rsfMRI analysis. Alternatively, the rsfMRI prediction analysis could incorporate the clinical variables (page 18) to show that rsfMRI truly explains additional variance over and above the clinical variables.

Minor:

Recent work regarding the prediction of pain-related traits from rsfMRI in adults could be referenced in the discussion (e.g. Yuan et al. 2019 and Spisak et al. 2020)

Reviewer #1 comments (blue) and author reply (black)

The authors purport to reveal brain mechanisms for noxious stimuli in infants. A fixed force pin prick stimulus related brain activity is studied in 18 infants. Responses are then related to resting state network properties with in and across independent data set. Moreover, the responses are related to white matter myelination properties.

The data analysis seem rigorous and observe activity seems genuinely a BOLD response. The authors extensively and comprehensively address worries that the collected signal could be artifactual.

Similarly relating these responses to resting state and to DTI parameters are properly executed.

However, the paper suffers from over-interpretation and this is a major flaw.

Given the study has no control stimulus to which observed brain activity can be contrasted with, we really have no idea what the observed brain activity responses correspond to. Moreover, it is clear that the activity is spread throughout the brain, and in 3 subjects responses are negative. Yet, the authors interpret all of this as nociceptive and pain related brain activity. Far simpler interpretation would be that these are simply arousal signals. The infant is exhibiting a generalized excitation. Or in the case of the 3 where negative responses are seen, they are showing diminished arousal. All the data the authors present are consistent and interpreted much more parsimoniously within such a framing. Otherwise the paper remains pure speculation.

As it stands, the manuscript's title, abstract, introduction and most of the discussion need complete reorganization.

We would like to thank Reviewer 1 for their thoughtful review. We are pleased that our analyses were considered to be rigorous, comprehensive and properly executed but disagree that our paper suffers from over-interpretation. We address this major concern by editing the manuscript to outline and defend four key points:

1. The use of a baseline reference is appropriate to study neonatal nociception
2. The neonates' PinPrick-evoked brain activity is localised to nociception-relevant brain regions
3. The neonates' PinPrick-evoked brain activity represents noxious-evoked/nociception-related brain activity
4. The neonates' PinPrick-evoked brain activity is not best described as arousal.

1. The use of a baseline reference is appropriate to study neonatal nociception

“Given the study has no control stimulus to which observed brain activity can be contrasted with, we really have no idea what the observed brain activity responses correspond to.”

Pain is a multidimensional phenomenon^{1,2}. In our study, we examined changes in neonates' noxious-evoked brain activity relative to their baseline brain activity (i.e. we used a baseline reference) as opposed to a control stimulus, because we are interested in the full multidimensional experience of the neonates, without subtracting any sub-component through use of a control stimulus. The study aim was not to disambiguate the psychological or computational subcomponents of nociception-related brain activity, but to understand the functional and structural bases for inter-individual variability in this noxious-evoked brain response. In our study, this is accomplished using a baseline reference and describing the noxious-evoked activity as multidimensional.

Our nociception paradigm and data analyses were designed with clinical translation in mind, due to the urgent need for improvements in understanding of neonatal pain assessment and treatment, as outlined in the opening of our paper:

“Neonates routinely undergo numerous painful procedures as part of standard clinical care shortly after birth during their stay in hospital³. Their lack of verbal communication, brief extra-uterine medical history, and ambiguity in the behavioural and physiological responses that underpin infant pain scales⁴, lead to a high degree of uncertainty in clinical decision-making related to the treatment of neonatal pain.” – page 3 paragraph 1.

We have edited the Discussion to clarify the rationale for and value in researching the full multidimensional noxious-evoked responses of neonates:

“We expect our experimental noxious stimulus evokes a multidimensional response profile in the neonatal brain, and sources of response variation may include variability in sensory discriminative aspects such as perceived stimulus intensity, cognitive aspects such as degree of arousal, salience, and attention, and potentially emotional aspects such as intensity of negative emotional valence^{1,2}. We focused on this overall multidimensional response assessment approach to facilitate harmonisation with existing validated multidimensional infant clinical pain assessment tools, such as the widely used PIPP-R scale (Premature Infant Pain Profile Revised)⁵⁻⁷ which integrates across multiple pain-relevant behavioural and physiological response features to provide a reliable measure of this complex phenomenon. Due to the subjective nature of pain and the non-verbal nature of neonates, facilitating this cross-modality integration is vital for mitigating limitations of each individual objective assessment approach^{8,9}.” – page 11 paragraph 3.

We appreciate that using a baseline reference does limit interpretability: without a control stimulus the evoked activity cannot be attributed to any specific aspect or dimension of nociception. However, in our paper, we do not make any interpretation of the evoked activity that attributes the activity to a single (experiential or computational) dimension or to any subset of dimensions. We have added the following Discussion to clarify this limitation, and we mention how limitations in interpretability can be addressed in future work:

“Improving biological interpretability of the functional and microstructural features included in this study will be required to fully appreciate the neurophysiological basis for the observed individual variability in nociceptive processing. Determining which aspects of the multidimensional noxious-evoked responses are predictable from resting-state activity would be valuable. However, designing complex experiments in neonates will be a major challenge. A tactic taken in a recent study involved applying multi-region fMRI pain signatures that capture distinct components of pain and negative affect in adults¹⁰⁻¹⁴ to neonates in order to disambiguate sensory-discriminative and cognitive components of the neonatal pain response¹⁵. This study puts forward a novel analysis-based approach to exploring the multifaceted neonatal response to noxious stimulation, which will complement future advances in experimental protocols.” – page 17 paragraph 3.

2. The neonates' PinPrick-evoked brain activity is localised to nociception-relevant brain regions

“Moreover, it is clear that the activity is spread throughout the brain”

We appreciate that it was not clear in our original submission that the PinPrick-evoked activity occurs in nociception-relevant brain regions, and we thank the reviewer for raising this point. We realise that in Figure 1 of our main text, which displays *unthresholded* evoked activity in a single brain slice, it is not possible to appreciate that the brain activity is localised to a limited set of functionally-relevant regions. The slice had been selected because it captured a high number of functionally relevant brain regions and displayed the wide range of

individual variability in PinPrick-evoked activity. Using unthresholded maps to visualise this variability is required to demonstrate the range of evoked response amplitudes from negative to negligible to positive amplitudes, without introducing the issues inherent to application of arbitrary thresholds.

We have conducted additional analysis and present new data in Supplementary Figure 1, reproduced below. These results show that the overall spatial pattern of PinPrick-evoked brain activity is not spread throughout the brain, but is localised to a specific subset of functionally relevant brain regions. This includes the pre- and post-central gyri, the superior parietal and paracentral lobules, insular and opercular cortices, and the thalamus¹⁶. We have added the following to our Results and Discussion sections:

“The noxious-evoked response was localised to functional brain regions classically considered part of the adult nociceptive pain system, including pre- and post-central gyri, opercular and insular cortices, and the thalamus¹⁶ (Supplementary Figure 1).” – page 6 paragraph 1.

“Here, we used a group-average response template to assess individual variability in evoked amplitudes. Thresholding this neonatal template, we confirmed the central importance of multiple brain regions classically considered part of the adult nociceptive pain system¹⁶, including the primary and secondary somatosensory cortices, thalamus, and insula (Supplementary Figure 1).” – page 11 paragraph 2.

Supplementary Figure 1: Group average (n=18) noxious-evoked brain activity. Regions in red are statistically significant clusters of activity. The activity map is aligned with the adult MNI template (grey-scale background) to facilitate labelling of functionally active regions using the probabilistic Harvard-Oxford Cortical Structural Atlas. Abbreviations: A,P,L,R = anterior, posterior, left, right; ic = insular cortex; oc = opercular cortex; pcl = paracentral lobule; pre/po = pre- / post-central gyri; spl = superior parietal lobule; thal = thalamus.

3. The neonates’ PinPrick-evoked brain activity is noxious-evoked/nociception-related brain activity

“Yet, the authors interpret all of this as nociceptive and pain related brain activity.”

Throughout the paper, we refer to the PinPrick-evoked activity as “noxious-evoked”, “nociception-related”, or similar terminology. We use the experimental noxious stimulus as a best available proxy for painful stimuli that allows us to ethically study neonatal nociception in the non-clinical experimental MRI setting. This point has been clarified in our Discussion:

“Our nociception paradigm involved applying a mild experimental sharp-touch stimulus to the neonate’s foot, evoking brain activity known to be similarly evoked by a range of tissue-damaging medical procedures, such as blood sampling, vaccinations, and cannulations^{17–19}. The tightly-controlled stimulus produces responses of lower amplitude than those of the clinical procedures and does not cause behavioural distress¹⁸, but activates A δ -fibres in the periphery and elicits noxious-evoked brain activity in

the cerebral cortex^{20–22}, making it a useful experimental tool to better understand neonatal pain processing.” – page 11 paragraph 2.

We believe it is not an “overinterpretation” or “pure speculation” to refer to the PinPrick-evoked brain activity as “noxious-evoked”, “nociception-related”, or similar terminology. The International Association for the Study of Pain (IASP) defines nociception as “the neural process of encoding noxious stimuli”²³. The PinPrick stimuli are regarded as nociceptive and are routinely used to study noxious-evoked brain activity in adults and infants using both fMRI^{20,24–27} and EEG^{17,18,28,29}. Therefore, consistent with the IASP definition and previous published literature, we use the term “noxious-evoked” or “nociception-related” when referring to the brain activity evoked by the PinPrick stimulus.

The additional analyses summarised by Supplementary Figure 1 (above) demonstrate that the noxious-evoked activity is well localised to brain regions classically considered part of the adult nociceptive pain system¹⁶. To further demonstrate the similarity of responses to adult pain-related activity, we performed additional analyses to directly compare our neonates’ PinPrick-evoked activity to adult activity patterns generated in Neurosynth (<https://www.neurosynth.org/>) using two terms most relevant to our study: “noxious” and “nociceptive”. We present the Neurosynth association test maps for these terms, as the association test maps display brain regions that are *preferentially* related to the term-of-interest, displaying voxels that are reported more often in articles that include the term-of-interest in their abstracts than articles that do not (Reply Figure 1 below). Due to our PinPrick stimulus being widely accepted as nociceptive, it is appropriate to cautiously use reverse inference to compare neonate and adult activity maps within closely matched nociception contexts (see^{15,20} for further examples of this approach). The top row of this figure (PinPrick-evoked activity) is the same result as displayed in Supplementary Figure 1. This adult-infant comparison highlights clear similarities between our neonate’s PinPrick-evoked activity map and adults’ meta-analysis “noxious” and “nociceptive” association test maps. These additional analyses further support our use of the terminology “noxious-evoked” and “nociception-related” when referring to the brain activity evoked by the PinPrick stimulus.

The methods used to generate the figures in Reply Figure 1 are as follows:

Neonate: This group-average (n=18) activity map (Reply Figure 1, rows 1) was generated using standard group-level voxelwise GLM analysis in FSL’s Randomise³⁰. Statistical significance was assessed non-parametrically using permutation testing with 10,000 permutations, a variance smoothing kernel of 6 mm FWHM due to limited degrees of freedom³¹, and cluster-based thresholding with a cluster-defining threshold of $z=3.1$ ($p=0.001$)³² and a cluster p-value of $p=0.05$.

Adult: These Neurosynth outputs are meta-analysis association test maps of activity associated with our two terms of interest: “noxious” and “nociceptive” (Reply Figure 1, rows 2-3). These association test maps are thresholded using default Neurosynth settings i.e. corrected using a false discovery rate (FDR) approach, with an expected FDR of 0.01. They are presented in Reply Figure 1 as directly downloaded from the Neurosynth website, without any further alterations.

Neonate, current study:
PinPrick-evoked activity

Adult, Neurosynth meta-analysis:
Noxious association test

Nociceptive association test

Arousal association test

Reply Figure 1: Row 1: The group-average ($n=18$) PinPrick-evoked activity observed in the neonate cohort of the current study. Evoked activity is localised to brain regions functionally relevant to the pain system¹⁶, and a labelled version of this activity can be found in Supplementary Figure 1. Row 2-3: Adult association test maps generate in Neurosynth highlighting brain regions that are preferentially related to the term “noxious” (row 2) and “nociceptive” (row 3). There is close correspondence between the neonatal map in Row 1 to the adult maps in Rows 2-3. Row 4: Adult association test maps generate in Neurosynth highlighting brain regions that are

preferentially related to the term “arousal”. There is very little overlap between the neonatal map in Row 1 to this adult “arousal” map.

4. The neonates’ PinPrick-evoked brain activity is not best described as arousal

“Far simpler interpretation would be that these are simply arousal signals. The infant is exhibiting a generalized excitation. Or in the case of the 3 where negative responses are seen, they are showing diminished arousal. All the data the authors present are consistent and interpreted much more parsimoniously within such a framing. Otherwise the paper remains pure speculation.”

In response to any form of stimulation, there is an element of arousal, and this will be one of many dimensions that comprise the response to noxious stimulation in our neonates. The question is thus whether our multidimensional interpretation is more appropriate than the unidimensional arousal interpretation. We do not believe attributing all PinPrick-evoked activity to arousal is warranted for two reasons.

First, we used Neurosynth to generate a meta-analysis association test map for adult “arousal” (Reply Figure 1, row 4). These brain regions that preferentially related to arousal do not overlap with regions of PinPrick-evoked activity in our neonatal nociception paradigm dataset. We do not suggest that this demonstrates a lack of arousal in our neonates in response to noxious stimulation. They will be aroused. But this incongruity between our neonate’s noxious-evoked activity patterns and adult association test “arousal” patterns does suggest that arousal should not be the preferred single descriptor of the evoked activity pattern. It is equally likely that variability in response amplitudes are driven by perceived intensity¹³ or salience³³. We have updated our Discussion to clarify this point and now explicitly mention arousal as a source of signal variability:

“We expect our experimental noxious stimulus evokes a multidimensional response profile in the neonatal brain, and sources of response variation may include variability in sensory discriminative aspects such as perceived stimulus intensity, cognitive aspects such as degree of arousal, salience, and attention, and potentially emotional aspects such as intensity of negative emotional valence^{1,2}.” – page 11 paragraph 3.

Second, adopting a uni-dimensional arousal interpretation is problematic in the three infants with negative BOLD responses to the noxious input. The reviewer suggests these responses represent diminished arousal, which would mean that these infants are more aroused at rest than when they are being actively stimulated with a sharp-touch nociceptive stimulus. This seems unlikely to us, as there is no theoretical or empirical evidence that suggests this type of response should happen in neonates. In the Discussion, we suggest that these negative BOLD responses may reflect immaturity in the haemodynamic response following increased neuronal activity - there are several sources of evidence for this interpretation outlined in the excellent review by Elizabeth Hillman’s group³⁴. We have updated our Discussion to include the following:

“In this maturity framework, our observed negative BOLD responses may then be explained by immaturity in neurovascular coupling mechanisms, such as vasoconstriction or insufficient functional hyperaemia to meet oxygen demands³⁴.” – page 14 paragraph 3.

In summary, we believe that interpreting the noxious-evoked activity as a multidimensional response to the noxious input is parsimonious and consistent with a body of published literature.

Reviewer #2 comments (blue) and author reply (black)

The main objective of this study is to better understand the inter-individual variability in pain perceptions in infants. The authors first aimed to predict functional MRI (fMRI) response maps to noxious stimulations from the

amplitude of resting-state functional networks. Second, they aimed to relate noxious responses to the individual white matter microstructure. This multimodal MRI study, combining task-based and resting-state fMRI and diffusion MRI, was performed in 18 newborns, and it also relied on analyses performed on data of more than 200 infants from the developing Human Connectome Project (dHCP).

The question raised by this study is highly interesting and still very little explored in newborns. The data and analyses carried out allow us to test the hypotheses put forward. The methodologies used are precise, adequate and quite detailed. In addition, the article is really enjoyable to read. Nevertheless, there are a few points that need to be clarified as detailed below.

We would like to thank the reviewer for this thoughtful and considerate review. We have substantially revised the manuscript in light of these comments and believe that the changes have vastly improved the presentation of our work.

Major concern 1: Exploration of "structure-function" relationships

My main concern is the way the "structure-function" relationships have been explored. Indeed, the prediction results based on resting-state networks are easy to understand and interpret since they put forward the somatomotor network (the interpretation is less obvious for the visual network). But the results for the diffusion MRI part are not completely clear to me. How could the 5 observed tracts be related to the pain system? What do the results mean compared with the reported resting-state networks? Furthermore, the analyses performed to address this question don't seem to me the most appropriate. Indeed, Supplementary Figure 7 (left) shows that the correlation coefficients are gradually decreasing, and considering only 5 out of 16 tracts is only a matter of statistical threshold considered (and correction considered for multiple comparisons?). Why not consider all the tracts in a prediction model, such as the one used with resting-state networks? Moreover, the relationships might be more a question of the overall microstructure of the white matter rather than the microstructure of specific tracts. The measurements for the individual tracts are probably highly correlated with the overall level of maturation of the white matter. Is it this level that is most relevant for predicting noxious responses? Or is it the deviations to this level for specific tracts? Why not then consider the first component of PCA on all tracts for correlations with noxious responses? The fact that the "structure-function" correlations between mean diffusivity values and responses are variable (i.e. with different correlation coefficients) between the two groups of newborns (Sup Fig 7) might support this hypothesis of low specificity of the tracts found.

Clarifying the relationship between the observed tracts and the pain system

"How could the 5 observed tracts be related to the pain system? What do the results mean compared with the reported resting-state networks?"

We appreciate that it was not clear in our original submission how the findings in our three MRI modalities interrelate, and we thank the reviewer for raising this point. We believe our white matter tract analysis identified five tracts that are of high functional relevance to pain and nociceptive processing, and intimately relate to our resting-state findings. This point has been clarified in our Discussion:

"Interestingly, this subset of tracts was intimately functionally related to both our nociception paradigm and the resting-state SMN. The superior thalamic radiations relay incoming nociceptive input from the somatosensory thalamus to primary somatomotor cortical regions, and the corticospinal tracts output motor signals, such as nocifensive actions. Together, these tracts form important structural connections for somatomotor aspects of pain processing³⁵. The anterior thalamic radiations and uncinat fasciculi connect limbic system regions involved in emotional and cognitive aspects of pain and nociceptive

processing³⁵. Finally, the forceps minor connects the medial and lateral frontal lobes of the left and right hemispheres, brain regions involved in higher-order integration and modulation of different aspects of pain signalling².” – page 14 paragraph 2.

Motivation for the function-based prediction model: feasibility for clinical translation

“Why not consider all the tracts in a prediction model, such as the one used with resting-state networks?”

Our nociception paradigm and data analyses were designed with clinical translation in mind, due to the urgent need for improvements in understanding of neonatal pain assessment and treatment, as outlined in the opening of our paper:

“Neonates routinely undergo numerous painful procedures as part of standard clinical care shortly after birth during their stay in hospital³. Their lack of verbal communication, brief extra-uterine medical history, and ambiguity in the behavioural and physiological responses that underpin infant pain scales⁴, lead to a high degree of uncertainty in clinical decision-making related to the treatment of neonatal pain.” – page 3 paragraph 1.

We have deliberately focused on the development of a resting-state function-based prediction model for neonates’ noxious-response amplitudes, as we believe there is greater clinical feasibility in translating a resting-state function-based approach to the cot-side compared to white matter microstructure-based approaches, because portable cot-side brain function imaging techniques (e.g. NIRS, EEG) are already readily available for use in post-natal wards and neonatal units. In our study, we focus on using structure-function associations to assess the role of temporally transient state effects versus temporally stable trait effects, as neonates’ noxious-response amplitudes will be influenced by both, and we believe models based on stable features are likely to be of greater clinical utility for neonatal pre-emptive pain management. We have clarified our approach in our paper:

“In this study, we demonstrate in a proof-of-principle manner that it is possible to predict a neonate’s cerebral noxious-evoked response from their pain-free resting-state activity. Translation of brain-function-based prediction models to common cot-side imaging techniques such as haemodynamic-based NIRS (near-infrared spectroscopy) or neurodynamic-based EEG (electroencephalography) would facilitate clinical feasibility, and progress the development of brain-based predictive approaches to tailoring pre-procedural or pre-operative decision-making regarding neonatal pain management strategies. We then demonstrated a structure-function relationship between noxious-response activity and white matter microstructure that accounted for over 20% of the total cross-subject variance in noxious-response amplitudes (Figure 5), suggesting the predictable evoked responses reflect relatively temporally stable trait effects. This feature of relative temporal stability (e.g. over a 24 hr window) will likely be valuable in developing predictive approaches with clinical utility to ensure insensitivity to transient state effects such as wakefulness, due to the inability to control a neonate’s sleep state.” – page 12 paragraph 2.

Both local and global microstructural maturity effects influence neonates’ noxious-response amplitudes

“Supplementary Figure 7 (left) shows that the correlation coefficients are gradually decreasing, and considering only 5 out of 16 tracts is only a matter of statistical threshold considered (and correction considered for multiple comparisons?). [...] Moreover, the relationships might be more a question of the overall microstructure of the white matter rather than the microstructure of specific tracts. The measurements for the individual tracts are probably highly correlated with the overall level of maturation of the white matter. Is it this level that is most relevant for predicting noxious responses? Or is it the deviations to this level for specific tracts? Why not then consider the first component of PCA on all tracts for correlations with noxious responses? The fact that the "structure-function" correlations between mean diffusivity values and responses are variable (i.e. with different

correlation coefficients) between the two groups of newborns (Sup Fig 7) might support this hypothesis of low specificity of the tracts found.”

We have clarified the rationale for focusing on the five specific tracts that survived statistical thresholding, despite the global structure-function association effects:

“Negative associations with MD and positive associations with FA existed for all tracts. However, we limited our data-driven hypotheses to the MD of these five specific tracts because they were the most robust subset of structure-function associations, and in addition to global effects, we expect some specificity to certain functionally relevant tracts.” – page 9 paragraph 2.

We have performed additional analyses to examine the topic of both local and global microstructural effects, and have substantially revised the text in both the main manuscript and supplementary information to outline our findings and explanations. The analyses and results are outlined in detail in the Supplementary Information section “Structure-function associations: global effects, non-uniformity, and nociception-relevance”. The key novel analyses and results are reproduced below in Supplementary Figures 7-8. In brief, we believe both local and global microstructural effects influence the neonates’ noxious-response amplitudes. Both of these effects fit with our original maturity interpretation that neonates with larger noxious-response amplitudes have more structurally and functionally mature brains.

The following text added to the Supplementary Information describes the analyses we performed to assess the explanatory value of the MD of our subset of 5 specific tracts (MD PC1 across the five tracts) versus the global MD effect (MD PC1 across all 16 tracts):

“Lastly, we examined the relative functional importance to noxious-response amplitudes of the MD of the five white matter tracts identified in the dHCP dataset exploratory analysis. The five tracts were the superior and anterior thalamic radiations, corticospinal tract, uncinate fasciculus, and forceps minor (Supplementary Figure 7, red box). As described in the main text, in our nociception-paradigm dataset, the first principal component of MD across these five tracts (MD PC1) explained over 20% of the variance in noxious-response amplitudes (Figure 5 bottom row; $r^2=0.21$). We compared the r^2 value for this specific 5-tract combination against every possible combination of the 16 tracts (i.e. MD PC1 r^2 values for all unique combinations) (Supplementary Figure 8). This distribution of MD PC1 r^2 values had a median of $r^2=0.17$ and a robust range (1st and 99th percentiles) of [0.08, 0.24]. The specific 5-tract MD PC1 $r^2=0.21$ featured in the 92.3th percentile. Additionally, variance explained by the MD PC1 across all 16 tracts lay close to the median with a $r^2=0.17$ at the 55.8th percentile. Thus, in the nociception paradigm dataset, like the dHCP dataset, the MD of this specific subset of five white matter tracts exhibited high explanatory value for inter-individual variability in neonates’ noxious-response amplitudes and outperformed the global MD metric assessed across all tracts which does not take into account the tract-to-tract variability in functional relevance.” – page 8 paragraph 3 (Supplementary Information).

These results are summarised and referenced in the main manuscript in the Results section.

“We also observed widespread negative associations with MD and positive association with FA for all 16 tracts (Supplementary Figure 7, top row), reproducing the global structure-function association effects observed in the dHCP dataset (Figure 4). Additionally, there was noticeable tract-to-tract variability in association strength in both datasets. Comparing this tract-to-tract variability between datasets, high similarity was observed for MD ($r=0.62$) (Supplementary Figure 7, bottom row), suggesting that tracts with strongest associations between MD and noxious-response amplitudes were relatively consistent between datasets. We compared the variance in noxious-response amplitudes explained by MD PC1 of these five specific tracts to that of the global signal (MD PC1 across all 16 tracts) as well as to all possible

combinations of the 16 tracts (Supplementary Figure 8). The specific 5-tract MD PC1 $r^2=0.21$ outperformed the global 16-tract MD PC1 $r^2=0.17$, and featured in the 92.3th percentile of the distribution of all possible tract combinations.” – page 10 paragraph 2.

Additionally, the following updated text has been added to our Discussion section to outline how the combination of both local and global structure-function associations fit with our maturity interpretation:

“As observed in fMRI-based resting-state studies, functional connectivity strength and activity amplitude in sensory, motor, and cognitive network all increase with age^{36,37}, with spatial and functional heterogeneity in timing and rate of development observed for both cortical structural and functional maturity^{38,39}. Similarly observed in white matter microstructure studies, white matter decreases in MD and increases in FA are global developmental trends observed during the perinatal period⁴⁰ and from infancy through to adulthood⁴¹. Analogous to cortical development, white matter maturation is both spatially and functionally heterogeneous in timing and rate of development^{38,41,42}. These global developmental increases in resting-state network amplitudes, increases in white matter FA, and decreases in white matter MD coupled with regional and functional variations could then explain our findings of global resting-state activity and white matter microstructure associations coupled with most robust associations identified in functionally relevant networks and tracts.” – page 15 paragraph 1.

Supplementary Figure 7: Cross-dataset similarities in structure-function associations. Top row: bar plots displaying the Pearson correlation coefficients between noxious-response amplitudes and microstructural features for each white matter tract. For each tract, the coefficients for both the dHCP dataset (grey) and nociception-paradigm (local) dataset (blue) are juxtaposed. Tracts are ordered consistently according to the dHCP MD correlation coefficients: from negative (top) to positive (bottom). The red box indicates the five dHCP correlation coefficients identified as statistically significant in our exploratory arm analyses. Bottom row: Correlation of

association strength for each tract across datasets. For each scatter plot, the dots indicate individual tracts with outliers highlighted in yellow. The black lines are the least squares fit lines, excluding outliers. The plots are divided into equal quadrants to illustrate polarity of the associations. Tracts with consistent polarities between datasets fall in the lower left and upper right quadrants. The global negative MD associations are visible as localisation of points to the lower left quadrant, and the global positive FA association are visible as localisation to the upper right quadrant. Abbreviations: ar = acoustic radiation; atr = anterior thalamic radiation; cgc = cingulate gyrus part of the cingulum; cgh = parahippocampal part of the cingulum; cst = corticospinal tract; fma = forceps major; fmi = forceps minor; for = fornix; ifo = inferior fronto-occipital fasciculus; ilf = inferior longitudinal fasciculus; mcp = middle cerebellar peduncle; ml = medial lemniscus; ptr = posterior thalamic radiation; slf = superior longitudinal fasciculus; str = superior thalamic radiation; unc = uncinata fasciculus.

Supplementary Figure 8: Explanatory value of functionally relevant tracts' MD relative to whole brain MD. For both histograms, the explanatory variable is the first principal component across the MD of white matter tracts (MD PC1). Pearson correlations between MD PC1 and noxious-response amplitudes are squared to calculate the proportion of variance explained (x-axis). The distributions are constructed from all possible unique combination of the 16 tracts. The red lines (MD5) are the results for the specific combination of five nociception-relevant tracts identified in our dHCP dataset exploratory analyses (Supplementary Figure 7, red box). The yellow lines (MD16) are the global MD PC1 results calculated across all 16 tracts. Left: These results from the dHCP dataset are biased due to the circularity in feature selection approach for MD5, but this plot demonstrates the effect-of-interest: the specific MD feature calculated from a subset of functionally relevant tracts has greater explanatory value than the global MD feature which ignores variability in tract relevance. Right: The specific 5-tract MD feature (MD5) exhibits greater explanatory value than the global 16-tract MD feature (MD16). Distribution median $r^2=0.165$; distribution robust range (1st and 99th percentiles) = [0.083, 0.238]; MD5 $r^2=0.210$ (92.3th percentile); MD16 $r^2=0.170$ (55.8th percentile).

Major concern 2: Cross-dataset similarities in diffusion features and data acquisition

Second, one point not mentioned in the article concerns the differences in the acquisition protocols (for resting-state fMRI and diffusion MRI) between the group of neonates with activation fMRI data and the dHCP cohort. How can this influence measurements and results? The authors show a good correlation between the resting-state networks identified in the 2 groups. Is it the same for the diffusion measurements of the tracts? For instance, the authors might evaluate whether the differences in diffusion parameters between tracts are similar across the 2 groups.

We have added a new table to the main text which compares data acquisition parameters between datasets (Table 3), and now cover this topic in our Discussion:

“These cross-dataset consistencies existed despite differences in scanner hardware and acquisition protocols (Table 3), and confirmed that our prediction model generalised well to the age-matched dHCP data, establishing initial indirect external model validation.” – page 12 paragraph 3.

This topic is now dealt with in detail in the Supplementary Information section “Cross-dataset similarities in functional and structural features”, and we have added new supplementary figure which compares frequency distributions of structure-function analysis features between datasets (Supplementary Figure 6). In this supplementary section, we state the following:

“Despite the differences in functional acquisition protocols and the reliance on predicted noxious-response amplitudes in the dHCP dataset, the frequency distributions of noxious-response amplitudes were very well aligned between the two datasets (Supplementary Figure 6, left). For each diffusion parameter, there was reasonably high consistency in frequency distributions between datasets, with greatest alignment visible for MD. Importantly, it was the structure-function associations that we attempted to replicate between datasets, so it is within-dataset relations between MRI modalities that is of primary interest. The minor cross-dataset variability in these functional and structural feature distributions (Supplementary Figure 6) is likely to have minimal influence on subject-to-subject variability within a dataset, and is thus unlikely to detrimentally impact comparisons of structure-function associations between datasets.

The close correspondence of both the functional and structural features (Supplementary Figure 6) and the structure-function associations between datasets (Figure 5 and Supplementary Figures 7-8) suggests that undesirable between-dataset inconsistencies has, at most, minor effects, and our resting-state prediction model generalises well to the age-matched dHCP data.” – page 6 paragraph 3 (Supplementary Information).

Table 3: Comparison of key data acquisition parameters between the nociception-paradigm (local) dataset and dHCP dataset for resting-state fMRI and dMRI data.

Parameter		Local dataset	dHCP dataset
Scanner	Model	Siemens Prisma	Philips Achieva
	Field strength	3T	3T
Head coil	Channel density	32-channel	32-channel
	Size	adult	neonatal
Resting-state	Echo time, TE (ms)	50	38
	Repetition time, TR (ms)	1300	392
	Multiband acceleration factor, MB	4	9
	Voxel size, isotropic (mm)	2	2.15
	Number of volumes	500	2300
Diffusion	Echo time, TE (ms)	73	90
	Repetition time, TR (ms)	2900	3800
	Multiband acceleration factor	3	4
	Voxel size, isotropic (mm)	1.75	1.5
	Number of volumes	163	300
	b-values (s/mm ²)	0, 500, 1000, 2000	0, 400, 1000, 2600
	Number of volumes/directions	20, 23, 50, 70	20, 64, 88, 128

	Gradient duration, δ (ms)	12.5	14
	Gradient separation, Δ (ms)	35.5	42.5

Supplementary Figure 6: Cross-dataset comparison of functional and structural features. For each frequency histogram, the dHCP dataset ($n=215$) distribution is in grey with the y-axis scale on the left, and the local nociception paradigm dataset ($n=17$) is in blue with the y-axis scale on the right. For the response amplitudes, the nociception paradigm dataset values are observed amplitudes, and the dHCP dataset values are predicted amplitudes. For the three white matter microstructural features, each infant contributed 16 values per parameter, an average parameter value for each of the 16 white matter tracts studied.

Minor concerns:

More minor comments are detailed below.

* The number of infants considered in the dHCP cohort needs to be clarified: is it 215 or 242? Are they all full-term newborns? Are their ages at birth and at MRI comparable to the 18 newborns of the study? In fact, resting-state networks and white matter microstructure differ between premature and full-term newborns.

The dHCP sample used in resting-state network identification ($n=242$) were previously analysed to identify resting-state networks as part of the dHCP. These neonates were all born and scanned at term age within the first few days of postnatal life, as so are fully age-matched to our nociception-paradigm dataset.

“To consider a network robust and suitable for inclusion in subsequent analyses, networks needed to be replicable across both the nociception-paradigm dataset ($n=18$) and an independent age-matched dHCP dataset ($n=242$) previously analysed using PFM⁴³.” – page 6 paragraph 3.

The dHCP sample used in the structure-function analysis ($n=215$) were prospectively selected to also be age-matched to our nociception paradigm dataset. This has been clarified in the Methods section “dHCP dataset sample selection”.

* The term “infant” should be replaced by “newborn” when referring to babies of less than 28 days post-natal age.

Thank you for this suggestion. Subjects included in our nociception-paradigm dataset and both dHCP datasets ($n=242$ and $n=251$) had postnatal ages under 28 days, so we have replaced the word “infant” with “neonate” throughout.

* When the authors describe the predictive results for clinical factors, they might emphasize that consequently it is not the youngest born or imaged newborns, or those with the lowest birth weight, who have the most immature responses.

This is an important point that we want to convey to the readers. We have revised the text and explicitly reiterated this point.

“Age is a good but imperfect proxy for neural maturational state assessment: two perfectly age-matched neonates would not be expected to be perfectly matched for maturational state, an extreme pathological example being cases of developmental dysmaturity⁴⁴. In this study, it was not the youngest born or imaged neonates, or those with the lowest birth weight, who had the smallest responses, suggesting individual variability in brain maturity is not fully captured by proxy indicators of neural maturity, such as age and brain volume (Supplementary Figure 5).” – page 15 paragraph 3.

* The authors might show the results for other diffusion parameters (anisotropy, mean kurtosis) in supplementary information.

We now provide the results for the matched analyses for FA and MK below (Reply Figure 2). Analogous to our MD analysis, the five tracts with strongest associations were used to derive the principal component white matter feature. And performing this analysis using the five specific tracts identified in our MD analysis results in even weaker associations for FA and MK. We could include these results at the request of the Editors or Reviewer, but are currently unsure that this figure aids the interpretation of our data.

Reply Figure 2: Structure-function associations between noxious-response amplitudes (y-axis) and white matter microstructure from five tracts using the first principal component, PC1 (x-axis). The dHCP sample (grey) includes n=215 infants, and the nociception-paradigm sample (blue) includes n=17 infants. Pearson correlation coefficients (r) and the associated p-value (p) are displayed in the title of each scatter plot.

* Why not consider parameters from NODDI model in addition to DTI and DKI?

The two major goals of our structure-function analyses were to understand the nature of neonates' noxious-response amplitudes regarding temporal stability and to validate our resting-state prediction model. These aims can be achieved with either representational models (DTI/DKI) or biophysical models (e.g. NODDI). The combination of MD, FA, and MK contain all the information, in terms of signal features, available to NODDI, so the information content is the same⁴⁵. We used DTI/DKI models firstly because, unlike NODDI, the model parameters are generated as part of the dHCP dMRI pipeline

(https://git.fmrib.ox.ac.uk/matteob/dHCP_neo_dMRI_pipeline_release)⁴⁰. Secondly, DTI/DKI likely offers greater sensitivity than biophysical models, while NODDI should have greater specificity⁴⁶. Given the modest size of our nociception-paradigm dMRI sample (n=17), we prioritised sensitivity over specificity. We have updated our Discussion section to highlight the value in adopting biophysical models such as NODDI in future work:

“Finally, we identified novel nociception-related structure-function associations using representational dMRI models (DTI/DKI). While the combination of MD, FA, and MK included in our study contain all the information, in terms of signal features, available to the biophysical model NODDI^{45,47}, future work applying both representational and biophysical models in complementary fashion would likely aid the biophysical interpretability of the microstructural phenomena underpinning our observed nociception-relevant structure-function associations.” – page 18 paragraph 2.

* Abstract

- The authors might precise which resting-state networks are the most important in the prediction, as well as which white matter tracts show correlations with noxious responses.

While we agree that this information could be valuable in the Abstract, the word limit prevents us from including the list of networks and tracts. We have updated our manuscript to specify the networks and tracts in each major section of the manuscript. We could include this information in the abstract at the request of the Editors or Reviewer, if exceeding the word limit to accomplish this would be allowable.

* Introduction

- I83-92: The order of sentences could be reviewed to improve comprehension.

This relevant section has been substantially revised. We have clarified this initial specific point, and the new relevant paragraph is as follows:

“Due to the large number of potential white matter features that could be studied and the lack of neonatal research into associations between noxious-evoked activity and underlying white matter microstructure that could guide the feature selection process, we adopted a two-part two-dataset approach, involving exploration of a range of structure-function associations in the dHCP dataset to formulate data-driven hypotheses, followed by independent confirmation of these hypotheses in our nociception-paradigm dataset. We use the dHCP dataset to explore possible structure-function relationships due to its larger sample size and thus greater statistical power. Given that the dHCP dataset does not include nociception-paradigm data, we generate a predicted noxious-response amplitude per neonate from their resting-state data using the prediction model originally trained in our local nociception-paradigm dataset. We focus on 16 white matter tracts previously used in a recent dHCP dMRI publication⁴⁰ and three tensor model parameters generated by the dHCP dMRI preprocessing pipeline (https://git.fmrib.ox.ac.uk/matteob/dHCP_neo_dMRI_pipeline_release): mean diffusivity, fractional

anisotropy, and mean kurtosis. Structure-function associations identified in this exploratory analysis were subsequently tested in our nociception-paradigm dataset, for which measured noxious-response amplitudes are available.” – page 4 paragraph 3.

- I93-94: This result is not that obvious.

It might be important to report which resting-state networks and which white matter tracts are used in the following analyses.

The global and local cross-dataset comparison results have been substantially revised and the relevant text updated in all sections. This has been thoroughly detailed above in reply to major concerns.

* Results

- I131: It's six networks instead of three.

This has been corrected:

“These included six sensory and motor networks (two visual, two auditory, and two somatomotor networks) and three cognitive networks (default mode, dorsal attention, and executive control networks).” – page 6 paragraph 3.

- I135: Is it 242 infants of the dHCP cohort for resting-state fMRI or 215?

The dHCP dataset for the resting-state network identification analysis is n=242. The dHCP dataset for the structure-function analysis is n=215. This has been clarified in the text.

“To consider a network robust and suitable for inclusion in subsequent analyses, networks needed to be replicable across both the nociception-paradigm dataset (n=18) and an independent age-matched dHCP dataset (n=242) previously analysed using PFM⁴³.” – page 6 paragraph 3.

“Using this model, predicted noxious-response amplitudes were generated for an age-matched dHCP sample (n=215) for which both resting-state and diffusion data were available.” – page 8 paragraph 3.

- I176: Here it would be interesting to better detail which networks contribute most to the prediction.

- I176: Beyond univariate correlation analyses, would it be possible to identify which features (i.e. networks) contribute most to the support vector regression model?

The somatomotor network is the single most predictive network. This analysis and results has been added to Supplementary Figure 5 and are referred to in our Results section:

“Examining the predictive value of each network individually, only the SMN exhibit notable predictive value, with a prediction performance ($R^2=0.60$) comparable to our multivariate prediction model performance ($R^2=0.64$) (Supplementary Figure 5, bottom row).” – page 8 paragraph 2.

Supplementary Figure 5: Resting-state coupling: global effects, non-uniformity, and nociception-relevance. Top: univariate correlations between neonates' noxious-response amplitudes and each of the 18 predictors in our three predictor categories: resting-state networks (resting-state), resting-state imaging confounds (confounds), and clinical variables. Correlation strengths are presented as proportion of variance explained (Pearson R^2). Positive correlations are in blue and negative in yellow. Of the 18 Pearson correlation tests, only VNop and SMN were statistically significant. Of the nine resting-state networks, eight exhibited positive correlations. Bottom: univariate predictions between neonates' noxious-response amplitudes and each of the 18 predictors. Prediction strengths are presented as proportion of variance explained (sums-of-squares R^2). The dashed grey lines indicate the prediction performance of each of the three multivariate prediction models for reference. Of all 18 predictors, only the SMN demonstrates meaningful predictive value, closely matching the overall performance of the multivariate resting-state prediction model. Abbreviations: VNm = medial visual network; VNop = occipital pole visual network; ANr = right auditory network; ANl = left auditory network; SMN = somatomotor network; DMN = default mode network; DAN = dorsal attention network; ECN = executive control network; Mot = head motion; CSF = cerebrospinal fluid signal amplitude; WM = white matter signal amplitude; PMA = postmenstrual age; GA = gestational age; PNA = postnatal age; BW = birth weight; TBV = total brain volume.

- I202-205: Here it would be interesting to detail which tracts are related to predicted noxious responses. How are these tracts consistent with the resting-state networks found previously?

We have included additional text to detail the relevant tracts and outline how they relate to the resting-state networks. This has also been detailed above in response to major concerns.

* Discussion

The discussion is interesting and well conducted. However, it could include a discussion of the specificity of the results found with regard to the correlations between structural and functional measures (see also previous points).

We have substantially updated our paper to cover this topic. This has been detailed above in response to major concerns.

* Methods

- I378: Subject information: How many newborns had to be included to get interpretable functional MRI data (task-based and resting-state) for 18?

These details have been added to the Methods section:

“Twenty one neonates had noxious-evoked and resting-state data collected. Subjects were excluded from analysis if (i) either scan session was not fully completed, in order to remove inter-subject variability in data quality related to scan length, and (ii) the vertex of the cerebral cortex left the scan field of view for more than 5% of the scan session, in order to ensure reliable data in this functionally relevant brain region. Using these criteria, three subjects were excluded resulting in a final sample of n=18 neonates.” – page 20 paragraph 2.

Are the fMRI data the same as those included in the study by Gokstan et al (2015)?

No, this is a new and fully independent dataset scanned in a different scanner with an improved acquisition protocol e.g. improved spatial and temporal resolutions as well as infant-optimised TE. A subset of these noxious-response data were included in a recent preprocessing optimisation paper ²⁵.

- I379: What is the number of directions for each b value? How many volumes for b=0 were acquired?

These acquisition details are now included in a new table, Table 3 (reproduced above).

- I423: What are the differences in acquisition protocols (for resting-state fMRI and diffusion MRI) between the newborn group of interest and the dHCP cohort? In particular, from which b values was the tensor model estimated for each group? Might this have an influence on the results? Would the repetition time (TR) have an influence on the characterization of resting-state networks?

These acquisition details are now included new table, Table 3. The MD and FA parameters were estimated from the b1000 data in both datasets (now mentioned in the Methods section). The influence of data acquisition on measurements and results has been detailed above in response to major concerns.

- I553: The authors should specify why they consider only 9 resting networks (this is said in the results section, but it would be useful to repeat it in the methods section).

The reasons for considering the nine resting-state networks are now also discussed in the Methods section:

“In brief, probabilistic functional mode (PFM) analysis using FSL’s PROFUMO ^{48,49} was run with a pre-specified dimensionality of 25, using the term-neonate double-gamma HRF ^{25,50} as the temporal prior. PROFUMO’s Bayesian model complexity penalties eliminated modes unsupported by the data, thus returning a number of group-level modes less than the pre-specified dimensionality. The data-determined dimensionality for the nociception-paradigm dataset was 11, nine of which corresponded to the dHCP dataset resting-state networks assessed using spatial correlation followed by visual confirmation.” – page 23 paragraph 1.

- I610: It might be appropriate to better introduce the reasoning behind this exploratory-confirmatory analysis approach in the introduction of the article.

The introduction of this topic has been updated for clarity:

“Due to the large number of potential white matter features that could be studied and the lack of neonatal research into associations between noxious-evoked activity and underlying white matter microstructure that could guide the feature selection process, we adopted a two-part two-dataset

approach, involving exploration of a range of structure-function associations in the dHCP dataset to formulate data-driven hypotheses, followed by independent confirmation of these hypotheses in our nociception-paradigm dataset. We use the dHCP dataset to explore possible structure-function relationships due to its larger sample size and thus greater statistical power. Given that the dHCP dataset does not include nociception-paradigm data, we generate a predicted noxious-response amplitude per neonate from their resting-state data using the prediction model originally trained in our local nociception-paradigm dataset. We focus on 16 white matter tracts previously used in a recent dHCP dMRI publication⁴⁰ and three tensor model parameters generated by the dHCP dMRI preprocessing pipeline (https://git.fmrib.ox.ac.uk/matteob/dHCP_neo_dMRI_pipeline_release): mean diffusivity, fractional anisotropy, and mean kurtosis. Structure-function associations identified in this exploratory analysis were subsequently tested in our nociception-paradigm dataset, for which measured noxious-response amplitudes are available.” – page 4 paragraph 3.

- I662: The authors should detail which tracts are of interest.

A detailed list of the tracts of interest has now been included in the Methods section:

“The 16 tracts used in our analyses were: acoustic radiation, anterior thalamic radiation, cingulate gyrus part of the cingulum, parahippocampal part of the cingulum, corticospinal tract, forceps major, forceps minor, fornix, inferior fronto-occipital fasciculus, inferior longitudinal fasciculus, middle cerebellar peduncle, medial lemniscus, posterior thalamic radiation, superior longitudinal fasciculus, superior thalamic radiation, uncinata fasciculus.” – page 26 paragraph 3.

- I702: Reporting similar correlation polarities doesn't seem to be significant evidence of similarity.

We agree that similar polarities on their own aren't strong evidence. However, we emphasise that this is one of a number of element of evidence that, when considered in combination, are complementary:

“The two hypotheses formulated in the dHCP dataset (see Results) were tested in the nociception-paradigm dataset. First, the polarity of the Pearson correlation coefficient between noxious-response amplitudes and individual tracts were qualitatively assessed. While similar correlation polarities on their own are not strong evidence, when considered in combination with other elements of evidence, this correlation polarity information provides useful complementary insight.” – page 28 paragraph 3.

* Figures

- Fig 1 and 2: The colour scale should be added.

Colour scales have been added to both Figures.

- Fig 3: Abbreviations could be detailed.

The Figure 3 caption has been edited to expand the abbreviations.

* Supplementary Information

- I117-118: I am not convinced by the observation that “a clear pattern emerges for both MD and FA”. What is the correlation, for each of the two parameters, between the correlation coefficients measured for the 16 tracts in the 2 groups?

We have substantially revised our presentation of these results, as outline in detail in our reply to major concerns. The correlation between datasets can be seen in Supplementary Figure 7 (reproduced above).

- I126-127: I agree with the observation that “a global effect common to all white matter tracts to varying degrees”. It would then seem to me relevant to consider other analyses, as mentioned at the beginning of the review.

We have thoroughly updated the sections dealing with these topics, as detailed above in response to major concerns.

- Sup Fig 2: It would be more appropriate to put the time scale in seconds, as in the previous figure. What do these numbers of outliers represent in percentage to the total number of images acquired for each subject? We have changed the time scale (x-axis) to seconds. In addition to the number of outliers, we have added the percentage of volumes containing an outlier in each of the nine network timeseries. See Supplementary Figure 3.

- Sup Fig 4: Are p values corrected for multiple comparisons performed?

This figure was not informative and has been removed. New results and figures requested by the reviewer have been added, as these present more useful data.

- Sup Fig 7: For mean diffusivity in the newborns group of interest, higher correlation coefficients are observed for many other tracts than the ones considered (as outlined by the dHCP cohort). What could that mean?

Due to the extensive consistencies in mean diffusivity results between the two datasets, these between-dataset inconsistencies could potentially be noise from any number of sources. As highlighted by the reviewer, there is variability in data acquisition protocols which might have some minor influences. Additionally, unavoidable limitations in accuracy of the resting-state prediction model will contribute further variability.

The authors might present the tracts in the same order for the different diffusion parameters (same order as for MD) to facilitate the comparisons across parameters.

In all bar plots of tract structure-function correlation, the tracts have been re-organised based on the MD structure-function correlation strength in the dHCP dataset.

- Sup Fig 9: It might be useful to clarify here the usefulness of these regions.

Text explaining the usefulness of these regions has been added to the Supplementary Figure 9 caption. This explanatory text matches that in the relevant text of the Supplementary Information in section "CSF and white matter regions-of-interest definition".

Reviewer #3 comments (blue) and author reply (black)

In this manuscript Baxter and colleagues investigate neurophysiological underpinnings of individual variability in noxious-evoked pain activity using resting state and white matter diffusion MRI in 18 healthy infants. The authors show that an infants' nociception related brain activity is associated with pain-free resting state activity and white matter microstructure. Further, the authors cross-validate the existence of resting state networks and their association with white matter microstructure in an independent data set of 215 infants from the human connectome project.

This paper addresses a scientifically and clinically very relevant topic. Knowledge regarding developmental aspects in the neurobiology of nociception and pain is sparse, but urgently needed given that early life exposure to noxious procedures is known to impact an infants developmental trajectory including the susceptibility for chronic diseases. This is a challenging investigation both from an experimental and methodological point of view and the authors should be applauded for acquiring this unique data set and their attempt to overcome inherent limitations of the small sample size by replicating parts of their results in data from the human connectome project.

Thank you for your positive comments, insightful review, and very helpful suggestions.

I have several comments that should be addressed

The authors should consider to replace the term pain by nociception at several places in the manuscript (e.g. “infant’s pain-related brain activity” in the abstract), or make clear that the use nociception related activity as a proxy of an infant’s painful experience.

We have reviewed our terminology related to the evoked brain activity and, where appropriate, have replaced the term “pain-related” with “noxious-evoked” throughout the manuscript. When pain is discussed we make it clear that we are using “noxious-evoked brain activity” as a proxy for pain-related activity. In the Abstract, we have changed “pain-related” to “noxious-evoked”.

In the Introduction and Discussion the authors allude to the potential clinical implications of their work. These statements should be either down-toned or elaborated in more concrete terms: how will multimodal MRI inform clinical decision making at an individual level? I am not a paediatrician but I doubt that MRI will ever become a routine diagnostic in infants.

We agree that it is unlikely MRI will become a routine clinical method for neonatal pain assessment. We have elaborated on the routes by which we believe our current findings could be translated to the clinical setting:

“In this study, we demonstrate in a proof-of-principle manner that it is possible to predict a neonate’s cerebral noxious-evoked response from their pain-free resting-state activity. Translation of brain-function-based prediction models to common cot-side imaging techniques such as haemodynamic-based NIRS (near-infrared spectroscopy) or neurodynamic-based EEG (electroencephalography) would facilitate clinical feasibility, and progress the development of brain-based predictive approaches to tailoring pre-procedural or pre-operative decision-making regarding neonatal pain management strategies. We then demonstrated a structure-function relationship between noxious-response activity and white matter microstructure that accounted for over 20% of the total cross-subject variance in noxious-response amplitudes (Figure 5), suggesting the predictable evoked responses reflect relatively temporally stable trait effects. This feature of relative temporal stability (e.g. over a 24 hr window) will likely be valuable in developing predictive approaches with clinical utility to ensure insensitivity to transient state effects such as wakefulness, due to the inability to control a neonate’s sleep state.” – page 12 paragraph 2.

While I can fully follow the rationale to replicate the identification of infants’ resting state and structural networks in an independent cohort, I have difficulty to assess the validity of the predicted amplitudes of noxious-evoked brain activity in the dataset from the human connectome project. Please comment. Also issues of circularity should be addressed if applicable.

We agree that we did not directly demonstrate the validity of the predicted noxious-response amplitudes in the dHCP dataset, as we do not have ground truth values with which to validate these predictions. We have revised our Supplementary Information to include a section that directly assesses the similarities between datasets of the functional and structural features of interest that form the basis of our structure-function association analysis. The Supplementary Information section is “Cross-dataset similarities in functional and structural features”, and the results are presented in Supplementary Figure 6 (reproduced above). As can be seen in this figure, there is very close correspondence between the distribution of measured response amplitudes from the nociception-paradigm dataset and the predicted response amplitudes from the dHCP dataset. Additionally, the several cross-dataset consistencies in structure-function associations (Figure 5 and Supplementary Figures 7-8) suggest the predicted amplitudes in the dHCP dataset reflect biologically similar neural features to that measured in the nociception-paradigm dataset.

We have updated our Discussion section to directly discuss this topic of the validity of the predicted amplitudes:

“We applied our prediction model to a large independent age-matched dHCP sample, generating predicted noxious-response amplitudes per neonate from their resting-state activity. The distribution of predicted response amplitudes in the dHCP closely matched the measured response amplitudes of our nociception-paradigm dataset, and several of the global and local structure-function association patterns initially identified in the dHCP dataset were subsequently confirmed in our nociception-paradigm dataset. These cross-dataset consistencies existed despite differences in scanner hardware and acquisition protocols (Table 3), and confirmed that our prediction model generalised well to the age-matched dHCP data, establishing initial indirect external model validation. These results demonstrate that it is possible to generate biologically meaningful noxious-response amplitudes in a large publicly available dataset lacking noxious-response brain activity data, but which features multimodal MRI data and a rich catalogue of interesting demographic variables. This will provide a platform for future MRI-based neuroscientific research, which could significantly advance our ability to study and understand the development of neonatal pain processing.” – page 12 paragraph 3.

In our updated Methods section, we now explicitly discuss the issue of circularity:

“Our structure-function analyses are divided into two sections: exploratory arm and confirmatory arm. It is necessary that the exploratory arm analyses are not informed by confirmatory arm structure-function associations, as this would introduce positive bias through circularity.” – page 27 paragraph 3.

Additionally, circular analyses performed in the dHCP dataset as part of the initial exploratory analysis is now explicitly highlighted in our updated Results section:

“Due to the presence of consistent structure-function correlation polarities across tracts identified in the dHCP dataset, variance across multiple tracts for a single dMRI parameter was pooled using principal component analysis (PCA). The first principal component across these tracts (MD PC1) was assessed for associations with noxious-response amplitudes. Similar to the single-tract correlation tests above, these multi-tract correlation tests were assessed using permutation tests with 10,000 permutations. In the dHCP dataset, these effect size and statistical significance measures are biased due to circularity in explanatory variable selection⁵¹, but the bias is restricted to the exploratory arm analyses.” – page 28 paragraph 2.

The authors use head motion, CSF amplitude and white matter amplitude as potential confounders of the resting state analysis. Please also consider age and total brain volume as a covariate in the rsfMRI analysis. Alternatively, the rsfMRI prediction analysis could incorporate the clinical variables (page 18) to show that rsfMRI truly explains additional variance over and above the clinical variables.

The clinical variables, which included age and total brain volume (TBV) were unable to predict noxious-response amplitudes in a multivariate prediction model, while the resting-state network amplitudes were (Figure 3 and Table 1). This indicates that the resting-state prediction model is explaining variance in the responses over and above that of the clinical variables.

We agree that age and TBV deserve further investigation into their predictive value. Due to the already large number of predictor and confound variables being estimated during cross validation training of our resting-state prediction model (nine predictors and three confound variables) relative to our modest sample size (n=18), we believe further increasing the number of predictors will cause issues due to increased uncertainty in estimation of predictor parameters. For this reason, and due to a similar line of questioning from Reviewer 2, we have instead taken the alternative approach of looking at the predictive value of each variable in isolation, which provides greater insight into the role of age and brain volume. These results are presented in Supplementary Figure 5 (reproduced above).

Similar to our multivariate prediction model trained on all six clinical variables, this univariate prediction model does not identify age, TBV, or any other clinical variable as noticeably predictive. However, we do not conclude that age and TBV are irrelevant to understanding infants' noxious-response amplitudes, as they would likely increase their predictive value when considering a wider age range.

“Age is a good but imperfect proxy for neural maturational state assessment: two perfectly age-matched neonates would not be expected to be perfectly matched for maturational state, an extreme pathological example being cases of developmental dysmaturity⁴⁴. In this study, it was not the youngest born or imaged neonates, or those with the lowest birth weight, who had the smallest responses, suggesting individual variability in brain maturity is not fully captured by proxy indicators of neural maturity, such as age and brain volume (Supplementary Figure 5). It is likely these features would increase their predictive value when considering a wider age range, so we do not conclude that age and brain volume are irrelevant to understanding neonates' noxious-response amplitudes. How well this maturity interpretation generalises to neonates outside of the studied age-range or to non-healthy non-normative populations, such as those born very prematurely, would be a highly informative route of enquiry.” – page 15 paragraph 3.

Minor:

Recent work regarding the prediction of pain-related traits from rsfMRI in adults could be referenced in the discussion (e.g. Yuan et al. 2019 and Spisak et al. 2020)

Thank you for drawing our attention to these recent and highly-relevant papers. We have now included reference to this work in our paper.

References

1. Tracey, I. & Mantyh, P. W. The cerebral signature for pain perception and its modulation. *Neuron* **55**, 377–391 (2007).
2. Wiech, K. Deconstructing the sensation of pain: The influence of cognitive processes on pain perception. *Science* **354**, 584–587 (2016).
3. Carbajal, R. *et al.* Epidemiology and treatment of painful procedures in neonates in intensive care units. *JAMA* **300**, 60–70 (2008).
4. Lee, G. & Stevens, B. Neonatal and infant pain assessment. in *Oxford Textbook of Paediatric Pain* 353–369 (Oxford University Press, 2013).
5. Stevens, B., Johnston, C., Taddio, A., Gibbins, S. & Yamada, J. The premature infant pain profile: evaluation 13 years after development. *Clin J Pain* **26**, 813–830 (2010).
6. Gibbins, S. *et al.* Validation of the Premature Infant Pain Profile-Revised (PIPP-R). *Early Human Development* **90**, 189–193 (2014).
7. Stevens, B. J. *et al.* The premature infant pain profile-revised (PIPP-R): initial validation and feasibility. *Clin J Pain* **30**, 238–243 (2014).
8. Worley, A., Fabrizi, L., Boyd, S. & Slater, R. Multi-modal pain measurements in infants. *J Neurosci Methods* **205**, 252–257 (2012).
9. Vaart, M. *et al.* Multimodal pain assessment improves discrimination between noxious and non-noxious stimuli in infants. *Paediatric and Neonatal Pain* **1**, 21–30 (2019).
10. Chang, L. J., Gianaros, P. J., Manuck, S. B., Krishnan, A. & Wager, T. D. A Sensitive and Specific Neural Signature for Picture-Induced Negative Affect. *PLoS Biol.* **13**, e1002180 (2015).
11. Kragel, P. A., Koban, L., Barrett, L. F. & Wager, T. D. Representation, Pattern Information, and Brain Signatures: From Neurons to Neuroimaging. *Neuron* **99**, 257–273 (2018).

12. Krishnan, A. *et al.* Somatic and vicarious pain are represented by dissociable multivariate brain patterns. *Elife* **5**, (2016).
13. Wager, T. D. *et al.* An fMRI-based neurologic signature of physical pain. *N. Engl. J. Med.* **368**, 1388–1397 (2013).
14. Woo, C.-W. *et al.* Separate neural representations for physical pain and social rejection. *Nat Commun* **5**, 5380 (2014).
15. Duff, E. P. *et al.* Inferring the infant pain experience: a translational fMRI-based signature study. *bioRxiv* 2020.04.01.998864 (2020) doi:10.1101/2020.04.01.998864.
16. Apkarian, A. V., Bushnell, M. C., Treede, R.-D. & Zubieta, J.-K. Human brain mechanisms of pain perception and regulation in health and disease. *Eur J Pain* **9**, 463–484 (2005).
17. Hartley, C. *et al.* Nociceptive brain activity as a measure of analgesic efficacy in infants. *Sci Transl Med* **9**, (2017).
18. Hartley, C. *et al.* The relationship between nociceptive brain activity, spinal reflex withdrawal and behaviour in newborn infants. *Scientific Reports* **5**, 12519 (2015).
19. Verriotis, M. *et al.* Cortical activity evoked by inoculation needle prick in infants up to one-year old. *Pain* **156**, 222–230 (2015).
20. Goksan, S. *et al.* fMRI reveals neural activity overlap between adult and infant pain. *Elife* **4**, (2015).
21. Slater, R. *et al.* Evoked potentials generated by noxious stimulation in the human infant brain. *Eur J Pain* **14**, 321–326 (2010).
22. Slater, R. *et al.* Cortical pain responses in human infants. *J. Neurosci.* **26**, 3662–3666 (2006).
23. IASP. IASP Terminology - IASP. <https://www.iasp-pain.org/Education/Content.aspx?ItemNumber=1698> (2020).
24. Baumgärtner, U. *et al.* Multiple somatotopic representations of heat and mechanical pain in the operculo-insular cortex: a high-resolution fMRI study. *J. Neurophysiol.* **104**, 2863–2872 (2010).
25. Baxter, L. *et al.* Optimising neonatal fMRI data analysis: Design and validation of an extended dHCP preprocessing pipeline to characterise noxious-evoked brain activity in infants. *Neuroimage* **186**, 286–300 (2019).
26. Goksan, S. *et al.* The influence of the descending pain modulatory system on infant pain-related brain activity. *Elife* **7**, (2018).
27. Williams, G. *et al.* Functional magnetic resonance imaging can be used to explore tactile and nociceptive processing in the infant brain. *Acta Paediatr* **104**, 158–166 (2015).
28. Green, G. *et al.* Behavioural discrimination of noxious stimuli in infants is dependent on brain maturation. *Pain* **160**, 493–500 (2019).
29. Iannetti, G. D., Baumgärtner, U., Tracey, I., Treede, R. D. & Magerl, W. Pinprick-evoked brain potentials: a novel tool to assess central sensitization of nociceptive pathways in humans. *J. Neurophysiol.* **110**, 1107–1116 (2013).
30. Winkler, A. M., Ridgway, G. R., Webster, M. A., Smith, S. M. & Nichols, T. E. Permutation inference for the general linear model. *Neuroimage* **92**, 381–397 (2014).
31. Holmes, A. P., Blair, R. C., Watson, J. D. & Ford, I. Nonparametric analysis of statistic images from functional mapping experiments. *J. Cereb. Blood Flow Metab.* **16**, 7–22 (1996).
32. Woo, C.-W., Krishnan, A. & Wager, T. D. Cluster-extent based thresholding in fMRI analyses: pitfalls and recommendations. *Neuroimage* **91**, 412–419 (2014).
33. Legrain, V., Iannetti, G. D., Plaghki, L. & Mouraux, A. The pain matrix reloaded: a salience detection system for the body. *Prog. Neurobiol.* **93**, 111–124 (2011).
34. Kozberg, M. & Hillman, E. Neurovascular coupling and energy metabolism in the developing brain. *Prog. Brain Res.* **225**, 213–242 (2016).

35. Lenz, F. A., Casey, K. L., Jones, E. G. & Willis, W. D. *The Human Pain System: Experimental and Clinical Perspectives*. (Cambridge University Press, 2010). doi:10.1017/CBO9780511770579.
36. Fitzgibbon, S. P. *et al.* The developing Human Connectome Project (dHCP) automated resting-state functional processing framework for newborn infants. *bioRxiv* 766030 (2019) doi:10.1101/766030.
37. Smyser, C. D. *et al.* Longitudinal analysis of neural network development in preterm infants. *Cereb. Cortex* **20**, 2852–2862 (2010).
38. Ouyang, M., Dubois, J., Yu, Q., Mukherjee, P. & Huang, H. Delineation of early brain development from fetuses to infants with diffusion MRI and beyond. *Neuroimage* **185**, 836–850 (2019).
39. Cao, M. *et al.* Early Development of Functional Network Segregation Revealed by Connectomic Analysis of the Preterm Human Brain. *Cereb. Cortex* **27**, 1949–1963 (2017).
40. Bastiani, M. *et al.* Automated processing pipeline for neonatal diffusion MRI in the developing Human Connectome Project. *Neuroimage* **185**, 750–763 (2019).
41. Dubois, J. *et al.* The early development of brain white matter: a review of imaging studies in fetuses, newborns and infants. *Neuroscience* **276**, 48–71 (2014).
42. Dubois, J. *et al.* Asynchrony of the early maturation of white matter bundles in healthy infants: quantitative landmarks revealed noninvasively by diffusion tensor imaging. *Hum Brain Mapp* **29**, 14–27 (2008).
43. Fitzgibbon, S. *et al.* The developing Human Connectome Project automated functional processing framework for neonates. *OHBM* (2018).
44. Tsuchida, T. N. *et al.* American clinical neurophysiology society standardized EEG terminology and categorization for the description of continuous EEG monitoring in neonates: report of the American Clinical Neurophysiology Society critical care monitoring committee. *J Clin Neurophysiol* **30**, 161–173 (2013).
45. Edwards, L. J., Pine, K. J., Ellerbrock, I., Weiskopf, N. & Mohammadi, S. NODDI-DTI: Estimating Neurite Orientation and Dispersion Parameters from a Diffusion Tensor in Healthy White Matter. *Front. Neurosci.* **11**, (2017).
46. Jelescu, I. O., Palombo, M., Bagnato, F. & Schilling, K. G. Challenges for biophysical modeling of microstructure. *Journal of Neuroscience Methods* **344**, 108861 (2020).
47. Zhang, H., Schneider, T., Wheeler-Kingshott, C. A. & Alexander, D. C. NODDI: practical in vivo neurite orientation dispersion and density imaging of the human brain. *Neuroimage* **61**, 1000–1016 (2012).
48. Harrison, S. J. *et al.* Modelling Subject Variability in the Spatial and Temporal Characteristics of Functional Modes. *bioRxiv* 544817 (2019) doi:10.1101/544817.
49. Harrison, S. J. *et al.* Large-scale probabilistic functional modes from resting state fMRI. *Neuroimage* **109**, 217–231 (2015).
50. Arichi, T. *et al.* Development of BOLD signal hemodynamic responses in the human brain. *Neuroimage* **63**, 663–673 (2012).
51. Kriegeskorte, N., Simmons, W. K., Bellgowan, P. S. F. & Baker, C. I. Circular analysis in systems neuroscience: the dangers of double dipping. *Nat. Neurosci.* **12**, 535–540 (2009).

Reviewer #1 (Remarks to the Author):

This revised manuscript claims to identify "neural features that could be used to predict neonatal pain responses". This is a novel and strong claim. However the conclusion is not justified by the evidence provided. The topic is important and clinically relevant and if the conclusion was justified it would be an important step forward.

Unfortunately, most of the initial criticisms raised by 2 reviewers remain inadequately addressed. The authors' responses to these criticisms repeatedly refer back to the notion that this is an important topic therefore we should accept the data even when the results remain minimally interpretable. I would like to point out that my position is the opposite. Given the clinically important topic and its potential for clinical use, it is imperative that the data are properly interpreted and adequately understood. Unfortunately, the paper as it stands falls far short of convincing me on these points.

The main weakness remains, as pointed out previously, lack of any adequate controls as well as lack of any clinical parameters that would justify that observed brain responses are related to nociception. The authors argue that the simple fact that the stimuli are noxious in the adults is sufficient to accept any related brain activity as a nociceptive signal. This is circular logic and unconvincing. Along this line of argument they now provide additional group average maps and indicate that the map has some similarity to meta-maps for nociception in the adult. Close examination of these maps shows that they are at best 50% similar, leaving one to wonder what we are really learning. Is the difference due to differences between adults and neonates or simply due to a large number of other potential interpretations. The important missing information is whether observed brain activity is differentially signaling nociception. In the absence of any control tasks we remain in the dark as to understanding what we are observing. Very simply, this paper would have a very different title and discussion if e.g. they also tested brain activity for gentle touch or for the voice of the mother or for the sound of hand clapping. One simply does not know whether the observed brain activity is a detection of an event in the environment or something more than that.

The rest of the data presented in the manuscript would all remain logical and fully consistent even if the activity interpretation is completely changed. For example, presence of resting state correspondences across groups and their relationship with stimulus-evoked activity signal size as well as the relationship with white matter properties all point to the concept that evoked activity is grounded on brain resting state properties and white matter development. None of these results however provide any additional evidence that the observed signal is in fact related to nociception, and thus has any predictive value of pain perception in neonates.

Simple biological correlates may have strengthened their argument but none is provided. The methods section mentions that heart rate and blood oxygenation was collected. One would be curious if these were correlated to the brain stimulus-response magnitude. Similarly, was there any evidence of reflexive responses to the stimulus and did these relate to brain-evoked responses.

The 3 cases where brain activity was negative raises many questions. The authors interpret them as a hemodynamic decoupling but this is pure speculation.

Overall, the general topic of nociception, pain, salience and their interrelationships, even in the adult, remains a complicated and contentious topic. Conclusions regarding pain in the neonate would require far better controlled studies.

Reviewer #2 (Remarks to the Author):

The authors answered the majority of my questions. I have only minor comments to make.

- To give a comprehensive overview of the study, an additional figure could show a flow chart of the successive analysis steps (described in the results section), on the different groups of subjects (with their respective number) and based on the different MRI modalities (nociception paradigm fMRI, resting-state fMRI, diffusion MRI). This would really help to understand the analyses performed, and how they fit together (e.g. for the dHCP database, resting-state networks have been first estimated on N=242 newborns, while only N=215 are included in the correlation analyses with the diffusion MRI).

- P26: The authors could specify the duration of the total acquisition protocol, as well as the duration for the fMRI protocol with the nociception paradigm, and for the diffusion MRI protocol.

- Table 2: is the order of the subjects the same as in Figure 1? That would be relevant and to be specified.
- Supplementary Figure 6: For the histograms of response amplitude, wouldn't it be $n=18$ for the local dataset?

Reviewer #3 (Remarks to the Author):

The authors have thoroughly addressed my previous comments and concerns in the rebuttal letter and the revised version of the manuscript. I have no additional comments.

Summary of major changes to manuscript

Revised Introduction to provide a better understanding of the background literature that links the observed brain responses to nociception

To better contextualise and motivate this work, we have revised the Introduction. We have clarified why we used MRI to measure neonatal brain responses to noxious stimulation and how these responses relate to clinical observations and behavioural measures of pain.

“In experimental settings, a multitude of complementary behavioural, physiological, and neural measures are used in an attempt to quantify neonatal pain, with a high degree of individual variability observed within all modalities¹⁻⁴. Due to the neural origin of pain and the recent feasibility of collecting multiple high-quality MRI imaging modalities of the neonatal brain within a single scan session, we used a multimodal MRI approach to investigate the neurophysiological basis for individual variability in neonates' blood oxygen level dependent (BOLD) responses to noxious stimulation. Our noxious stimulation paradigm involved applying a mild experimental sharp-touch stimulus (PinPrick Stimulator, MRC Systems) to the neonate's foot, evoking brain activity known to be similarly evoked by a range of tissue-damaging medical procedures, such as blood sampling, vaccinations, and cannulations^{1,5,6}. The pinprick stimulus produces responses of lower amplitude than those of clinical procedures and does not cause behavioural distress¹, but activates A-fibre nociceptors in the periphery⁷ and elicits noxious-evoked brain activity in the cerebral cortex⁸⁻¹¹, making it a useful experimental tool to better understand neonatal pain processing.” – page 3 paragraph 2.

We describe the previously reported relationships between the noxious-evoked cerebral response and pain and nociception, and provide useful definitions for both pain and nociception – in accordance with the International Association for the Study of Pain¹². We also describe why a characterisation of the overall noxious-evoked response properties is a pertinent and accessible measure for understanding infant pain:

“Due to the emergent and multifaceted nature of pain^{13,14}, we focus on assessing the overall BOLD response amplitude. While this BOLD response neither directly reflects nociception, the neural process of encoding noxious stimuli¹², nor pain perception, the unpleasant sensory and emotional subjective experience¹², it is a pertinent and accessible feature of central importance to understanding neonates' neural responses to noxious input and the neurophysiology of the early developing pain system. This overall noxious-evoked BOLD response pattern resembles that of adults⁸ and expresses the adult Neurologic Pain Signature (NPS)¹⁵, a multivariate fMRI signature predictive of adult verbal reports of physical pain. The overall response captures inter-individual variability in the multidimensional noxious-evoked activity, which is tightly linked to the pre-stimulation functional status of the descending pain modulatory system¹⁶ and is likely to be driven by variability in sensory-discriminative, cognitive, and emotional aspects.

Furthermore, a holistic multidimensional noxious-evoked brain response metric reflects and should facilitate future harmonisation with existing validated multidimensional infant clinical pain assessment tools, such as the widely used PIPP-R scale (Premature Infant Pain Profile Revised)¹⁷⁻¹⁹, which integrates across multiple pain-relevant behavioural and autonomic response features to provide a reliable overall measure of this complex phenomenon²⁰. Due to the subjective nature of pain and non-verbal nature of neonates, having to rely on objectively measured noxious-evoked response features for infant pain measurement is a major challenge for the field of infant pain research, and thus facilitating this cross-modality integration is vital for mitigating limitations of each individual objective assessment approach^{21,22}.” – page 4 paragraphs 2-3.

Revising our terminology, acknowledging the study limitations and toning down our conclusions

We have gone through the manuscript to fully and appropriately standardise our use of the relevant terminology, most importantly the terms “noxious-evoked”, “nociception”, and “pain”. We acknowledge that our previous use of these terms unintendedly inflated the specificity of our claims. Here, we highlight two key examples of this. First,

referring to the brain activity evoked by the 128mN pinprick stimulus as “nociceptive” or “nociception-related” was too strong as it explicitly specifies nociception. We now more appropriately refer to the evoked activity as “noxious-evoked” due to it being activity evoked by a noxious stimulus. Second, referring to our experimental design as a “nociception paradigm” was again too specific to nociception. We now describe our paradigm as a “noxious stimulation paradigm” as the paradigm unambiguously involves noxious stimulation, which may be used to study nociception or pain or both, depending on the focus of the research question. These terminology changes that tone down our conclusions regarding the specificity of our findings to nociception can be seen throughout our manuscript.

The overall brain response to peripheral stimulation will be a complex combination of incoming signals specific to the peripheral receptors activated by the sensory stimulus, as well as higher-level neural events such as attention, salience, and arousal that lack specificity to the stimulus modality. In order to examine the balance of influences of each contributing aspect to the overall stimulus-evoked brain activity, experimental designs that include appropriate control stimuli are required to decompose the measured response into its constituent components. Given the motivation for our interest in neonates’ holistic multidimensional noxious-evoked responses, we assessed overall noxious-evoked responses relative to baseline. This design thus precludes the use of control stimuli to decompose the overall noxious-evoked response and isolate nociceptive responses, signal features specific to the encoding of the noxious stimulus. In our updated Discussion section, we now acknowledge more clearly that the between-subject variation in noxious-evoked amplitudes likely reflects more than nociception, and discuss the potential contribution and challenges for control conditions to bring light to these questions. In pain research in general, even in adults, it is not trivial how to control for or disambiguate the different aspects of pain, how to remove all non-nociceptive aspects to unambiguously demonstrate nociceptive information encoded in the BOLD response, or whether non-nociceptive aspects should be removed if the interest is in pain rather than nociception:

*“Pain is a multidimensional phenomenon that transcends nociception with sensory, cognitive, and emotional components to response variability^{13,14}. **The observed inter-subject variability in response amplitudes assessed in the current study may be driven by sensory-discriminative aspects such as perceived stimulus intensity, cognitive aspects such as arousal or attention, or emotional aspects such as intensity of unpleasantness. The specific balance of processes contributing to the overall BOLD response is uncertain.** Decomposing measured neural responses to noxious stimuli into constituent components is an important but challenging task. Control stimuli that are matched to the noxious stimulus in all non-nociceptive aspects could help to isolate nociceptive contributions to the signal. However, these are difficult to achieve in adults, and even more so in infants, and it is arguably more pressing from a clinical perspective to understand the neurophysiological basis for the overall multifaceted response than an individual constituent component of the signal.” – page 15 paragraph 3.*

It is important to highlight that these limitations are not specific to the present study, and linking the noxious-evoked brain activity to clinical and biological correlates, such as behavioural reflexes, heart rate, and blood oxygenation, cannot solve this general specificity limitation. This important discussion point is now also explicitly covered in our Discussion section (see below).

Additional substantive analysis to strengthen the study conclusions

We have now introduced a series of new analyses that further strengthen our conclusions that the neonates’ overall noxious-evoked responses are pain-relevant signals:

“We augment the analysis of overall BOLD responses with a parallel study of the expression in this data of Neurosynth-derived templates²³ and adult pain signatures, providing insight into the processes contributing to the observed responses.” – page 5 paragraph 1.

These new analysis outputs have been incorporated into our Results section, which now demonstrate that:

- (i) In our group of neonates, significant correspondences exist between the overall neonatal patterns of noxious-evoked BOLD responses and template response patterns associated with adult pain, and no

equivalent correspondences exist for **a number of control activity patterns** (Figure 2bi and Table 1 T-test results).

- (ii) Inter-subject variability in overall noxious-evoked response amplitudes strongly correlates with the inter-subject variability in correspondences with adult pain templates, such that the stronger a neonate's BOLD response to the noxious stimulus, the more it correlates with both adult pain signatures (Figure 2bii-iii and Table 1 Correlation results). Additionally, no equivalent correlations were found for the "control" activity patterns (Supplementary Figure 9 and Table 1 Correlation results).
- (iii) Pain-related components of the overall noxious-evoked response can be predicted from resting state brain activity.

The following new text, figures, and tables from our updated Results and Supplementary Information sections now outlines these important analysis contributions, which significantly strengthen our conclusions that the noxious-evoked BOLD responses are pain-relevant signals:

"To gain insight into processes underlying the observed noxious-evoked activity, we assessed expression of two template maps that have been independently linked to pain in adults. Both the Neurosynth²³ pain association test map (<https://www.neurosynth.org/analyses/terms/pain>), derived from a search of a meta-analytic database of fMRI study activation co-ordinates for the keyword "pain", and the adult Neurologic Pain Signature (NPS)²⁴, which predicts variations in adult verbal reports of pain intensity, were significantly expressed (Figure 2bi and Table 1 T-test results). We also assessed expression of a number of related and control templates. As negative controls, we assessed the Neurosynth "visual" pattern as well as the adult social rejection pain signature²⁴, and found neither to be expressed. Of the sensory, cognitive, and emotional Neurosynth patterns assessed, only the "nociceptive" pattern was significantly expressed, while patterns for "attention", "unpleasant", "salience", and "arousal" were not (Figure 2bi and Table 1 T-test results).

In addition to group average template expression, we assessed associations between the inter-individual variability in overall noxious-evoked response amplitudes (regression coefficients) and the correspondence between neonates' noxious-evoked response maps and each template (correlation coefficients). Of the nine adult templates assessed, significant associations were only observed between noxious-evoked response amplitudes and adult template expression for NPS and Neurosynth Pain and Nociceptive templates (Figure 2ii-iii and Table 1 Correlation results). Thus, the larger the overall BOLD response amplitude to noxious stimulation, the closer the correspondence with adult pain and nociceptive signatures. Correlation results for all templates are presented in Supplementary Figure 9 and Table 1. Collectively, these adult template results support the interpretation that the neonates' overall noxious-evoked responses are pain-relevant signals." – page 8 paragraphs 2-3.

Figure 2: Noxious-evoked responses are pain-relevant signals. **a.** The thresholded group average noxious-evoked map displays *t*-statistics in statistically significant clusters. Activity is localised to regions classically considered part of the adult nociceptive pain system, including the pre- and post-central gyri (pre/po), paracentral and superior parietal lobules (pcl and spl), opercular and insular cortices (oc and ic), and thalamus (thal). **b. i.:** For each infant, expression of functional templates (x-axis) is assessed as whole-brain Pearson correlations between the template and neonates' noxious-evoked response maps (y-axis). Group average template expression was assessed using two-tailed *t*-tests ($n=18$). Grey and red dots represent the group mean correlation coefficient, with grey bars displaying 95% confidence intervals (CI). The thresholded noxious-evoked map (displayed in part a) is the positive control. Visual is the Neurosynth negative control, and Social Rejection Pain is the pain signature negative control. The Neurologic Pain Signature (NPS) and Neurosynth Pain and Nociceptive templates were significantly expressed in this group of neonates, while none of the negative controls or other Neurosynth templates were significantly expressed. **ii-iii.:** Using two-tailed Pearson correlation tests to assess inter-subject variability in noxious-evoked responses, strong associations exist between the overall noxious-evoked response amplitudes (regression parameters) and both NPS and Neurosynth Pain correspondences (correlation coefficients). The dashed grey line is the least squares fit. The stronger the neonatal BOLD response amplitude to the noxious stimulus, the closer the correspondence with both adult pain signatures. *T*-test and correlation test results for all templates are summarised in Table 1.

Table 1: Noxious-evoked responses are pain-relevant signals.

	Noxious-evoked	Visual	Pain	Nociceptive	Attention
T-test	4.68* (0.0007)	-0.51 (0.62)	3.85* (0.0022)	4.34* (0.0013)	1.88 (0.078)
Correlation	0.88* (0.0001)	0.18 (0.47)	0.89* (0.0001)	0.87* (0.0001)	0.28 (0.26)
	Unpleasant	Saliency	Arousal	NPS	Social
T-test	0.49 (0.66)	-0.15 (0.89)	-0.34 (0.76)	3.71* (0.0016)	0.58 (0.56)
Correlation	-0.33 (0.18)	-0.11 (0.66)	-0.32 (0.19)	0.77* (0.0002)	0.36 (0.14)

T-test results assess the group average presence or absence of template expression within the neonates' ($n=18$) noxious-evoked response maps. *T*-statistics and *p*-values are presented for each template. Correlation results assess the correspondence between the inter-individual variability in overall noxious-evoked response amplitudes and the correspondence between neonates' noxious-evoked response maps

and each template. Pearson correlation coefficients and p -values are presented for each template. P -values are presented in parentheses. * = statistically significant.

Supplementary Figure 9: Associations between inter-individual variability in noxious-evoked response amplitudes and functional template correspondence. In all plots, each blue dot is a single subject ($n=18$), the dashed grey line is the least-squares fit, the x-axis is the overall noxious-evoked response amplitude (Figure 1 scalar values), and the y-axis is the correspondence (Pearson correlation coefficient) between each neonate’s noxious-evoked response map (Figure 1 heat maps) and an adult functional template. Associations between noxious-evoked response amplitudes and adult functional template correspondence was assessed using two-tailed Pearson correlation tests. Statistical results for the nine adult templates are presented in Table 1. Of the nine adult templates assessed, only the NPS and Neurosynth Pain and Nociceptive templates had statistically significant associations, such that neonates with larger noxious-evoked response amplitude had noxious-evoked response maps exhibiting closer correspondence with adult signatures of pain and nociception.

Additionally, we demonstrate that these pain components of the overall noxious-evoked response can themselves be predicted from the resting state data, which significantly strengthens our conclusions that nociception-free resting-state brain activity can be used to predict neonates’ pain-relevant responses:

“Resting-state network amplitudes were also predictive of both the Neurosynth ($R^2=0.46$, $p=0.006$) and NPS ($R^2=0.42$, $p=0.013$) response magnitudes (Table 2).” – page 10 paragraph 2.

Table 2: Noxious-evoked response amplitude prediction performance.

Predictors	Responses	R ²	RMSE	R _{Sp}
Resting state	Overall	0.62* (0.003)	1.57* (0.003)	0.79* (0.001)
	Neurosynth pain	0.46* (0.006)	0.15* (0.006)	0.65* (0.01)
	NPS	0.42* (0.013)	0.025* (0.013)	0.62* (0.014)
Clinical variables	Overall	0.11 (0.25)	2.42 (0.25)	0.36 (0.19)
Confounds	Overall	0.081 (0.56)	2.46 (0.56)	0.14 (0.44)

Each row contains results for a specific set of predictors and responses. Each results column contains a prediction performance metric: R² = coefficient of determination (sums-of-squares formulation); RMSE = root mean squared error; R_{Sp} = Spearman's rank correlation coefficient. P-values are presented in parentheses. * = statistically significant.

The following text from our updated Discussion section details how these novel findings strengthen our conclusions that the noxious-evoked BOLD responses are pain-relevant signals:

“Here, we adopted a template-based approach to demonstrate that variability in noxious-evoked activity in neonates shows concordance with adult fMRI activity associated with pain and nociception, and demonstrate the expression of the NPS, which tracks adult verbal pain ratings¹⁵. These results provide important supporting evidence for the processes contributing to the noxious-evoked response, and gives us some confidence that the brain responses reflect a set of pain-related processes with some concordance with existing infant and adult pain measures.” – page 16 paragraph 3.

We have not included additional analyses to link the neonates' noxious-evoked brain activity to clinical and biological correlates, such as limb reflex, heart rate, and blood oxygenation. Technical and safety challenges exist that precluded the incorporation of EMG (electromyography) into the MRI environment to assess pain-relevant reflex behaviours, and the ethical requirement to use mild noxious stimuli for experimental study of the neonatal pain system results in unreliable estimates of inter-subject variability in autonomic responses (e.g. cardiovascular-related or respiratory-related effects) due to the small effect size associated with the pinprick stimulus. Moreover, clinical behavioural and autonomic/biological response measures cannot demonstrate the nociceptive nature of the noxious-evoked brain activity. They are neither specific to pain nor nociception, nor other non-nociceptive components of pain.

*“Several infant clinical pain scales are multidimensional in nature²⁵, such as the PIPP-R scale which takes into account behavioural and autonomic responses. Previous results from our lab demonstrate that pinprick-evoked brain responses measured with EEG (electroencephalography) correlate with both reflex behaviour assessed with EMG (electromyography)¹ and autonomic heart rate responses⁵. However, technical and safety challenges in incorporating EMG into the MRI environment and unreliable estimates of inter-subject variability in autonomic responses (due to the small effect size associated with the pinprick stimulus), precluded linking the BOLD response amplitudes to these behavioural and autonomic dimensions at the individual level in the current study. **But it is important to note that these behavioural and autonomic responses lack specificity to pain dimensions^{25,26}, and thus cannot be used to decompose the noxious-evoked BOLD response into sensory-discriminative, cognitive, or emotional contributions, or demonstrate that evoked BOLD responses differentially reflect nociception.**” – page 16 paragraph 2.*

Reviewer #1 Comments (blue) and author reply (black):

Thank you for your detailed comments and support in ensuring that our data are clearly described and accurately interpreted. In light of your comments, specific changes to our manuscript were requested, which we have addressed in a substantial revision and outlined above. Additional specific points raised in your review are addressed in turn.

This revised manuscript claims to identify "neural features the could be used to predict neonatal pain responses". This is a novel and strong claim. However the conclusion is not justified by the evidence provided. The optic is important and clinically relevant and if the conclusion was justified it would be an important step forward.

We agree with Reviewer 1 that this sentence in the Abstract is open to misinterpretation and should be rephrased. We want to ensure our message is communicated unambiguously and we are grateful to the reviewer for highlighting this point. The sentence has now been rephrased to make it clear that we demonstrate that specific functional features of the neonatal brain can be used to predict ‘*responses to noxious stimulation*’, but this cannot be interpreted as a measure of pain perception.

We now state:

“This study in healthy neonates demonstrates that noxious-evoked brain activity is tightly coupled to both resting-state activity and white matter microstructure, that neural features can be used to predict responses to noxious stimulation, and that the dHCP dataset could be utilised for future exploratory research of early life pain system neurophysiology.” – page 2 paragraph 1.

In the Introduction we have further clarified and toned-down our language. We make it clear that characterising the neonates resting-state activity and white matter microstructure does not enable us to predict nociception or subjective pain experience, but rather these features can be used to predict their cerebral responses to noxious stimulation.

“Due to the emergent and multifaceted nature of pain^{13,14}, we focus on assessing the overall BOLD response amplitude. While this BOLD response neither directly reflects nociception, the neural process of encoding noxious stimuli¹², nor pain perception, the unpleasant sensory and emotional subjective experience¹², it is a pertinent and accessible feature of central importance to understanding neonates’ neural responses to noxious input and the neurophysiology of the early developing pain system. [...] Due to the subjective nature of pain and non-verbal nature of neonates, having to rely on objectively measured noxious-evoked response features for infant pain measurement is a major challenge for the field of infant pain research, and thus facilitating this cross-modality integration is vital for mitigating limitations of each individual objective assessment approach^{21,22}.” – page 4 paragraphs 2-3.

Unfortunately, most of the initial criticisms raised by 2 reviewers remain inadequately addressed. The authors responses to these criticisms repeatedly refer back to the notion that this is an important topic therefore we should accept the data even when the results remain minimally interpretable. I would like to point out that my position is the opposite. Given the clinically important topic and its potential for clinical use, it is imperative that the data are properly interpreted and adequately understood. Unfortunately, the paper as it stand falls far short of convincing me on these points.

We regret that our initial responses were interpreted to imply that we consider our data should be accepted by virtue of the importance of the topic – we certainly do not consider this to be the case! We hope that the revised manuscript, careful rephrasing, and additional analyses clarify our claims and the context in which this data can be interpreted.

The main weakness remains, as pointed previously, lack of any adequate controls as well as lack of any clinical parameters that would justify that observed brain responses are related to nociception.

It is unfortunate that control stimuli were not possible to acquire, but we hope that our additional analyses, which include positive and negative control spatial patterns of activity, and careful discussion of this point address the primary concerns of the reviewer (see Summary of major changes to manuscript). We make clear that we are not primarily targeting nociception, but a marker of the overall noxious-evoked response. While we do not claim to identify the specific balance of processes contributing to the over BOLD response, our additional analyses demonstrate clearly that the measured responses reflect key elements of the processes underlying pain. It is unlikely, given this evidence, that the responses simply reflect processes with no nociceptive component.

Notably, we now demonstrate that our neonates' noxious-evoked response maps differentially express patterns of brain activity associated with two independently constructed adult pain templates (Figure 2bi and Table 1 T-test results). The inter-individual variability in neonates' noxious-evoked response amplitudes covary with the correspondence between neonates' response maps and both adult pain signatures (Figure 2bii-iii and Table 1 Correlation results). Additionally, we tested negative control adult template patterns of activity as well as several templates associated with cognitive and emotional pain dimensions. None of these negative control and cognitive and emotional adult templates were significantly expressed or covaried with the neonates' overall noxious-evoked response amplitudes (Figure 2bi, Supplementary Figure 9, Table 1 T-test and Correlation results).

The authors argue that the simple fact that the stimuli are noxious in the adults is sufficient to accept any related brain activity as a nociceptive signal. This is circular logic and unconvincing.

We now make it clearer that we do not claim that all of the elicited activity reflects a nociceptive signal. Our new analyses provide strong evidence that the fMRI responses reflect, in part, nociceptive processes. Future work with control stimuli would provide additional insight into the contributing processes to the noxious-evoked responses, and whether other stimuli might similarly be predicted from resting state data. However, even in adults, control stimuli provide only limited additional ability to distinguish elements of the pain response, a topic we now explicitly cover in our Discussion. Our approach to use signatures derived from verbal reports and other experiments in adult populations provides more direct evidence that pain-related processes contribute to the cross-subject variability that we are analysing.

Along this line of argument they now provide additional group average maps and indicate that the map has some similarity to meta-maps for nociception in the adult. Close examination of these maps shows that they are at best 50% similar, leaving one to wonder what we are really learning. Is the difference due to differences between adults and neonates or simply due to a large number of other potential interpretations.

We accept that the qualitative map comparisons presented in the previous manuscript did not present a compelling case for the relation between the infant noxious-evoked responses and adult meta-maps for nociception. We now provide a more quantitative and extensive analysis of these maps, expanding our assessment to maps reflecting a variety of different components of the pain response as well as some control maps. In addition, we assess the Neurologic Pain Signature, which has been developed to predict verbal reports of pain intensity in adults. Our new analyses and results are detailed above in our Summary of major changes to manuscript – see Figure 2b, Supplementary Figure 9 and Table 1.

These results clearly show that the infant responses reliably correspond to pain-related maps in adults. In particular, non-nociceptive aspects such as arousal or salience do not predominate, and reasonable negative control maps (patterns associated with visual-evoked activity and social rejection pain) are not expressed.

The important lacking information is whether observed brain activity is differentially signaling nociception. In the absence of any control tasks we remain in the dark as to understanding what we are observing. Very simply, this paper would have a very different title and discussion if e.g. they also tested brain activity for gentle touch or for the voice of the mother or for the sound of hand clapping. One simply does not know whether the observed brain activity is a detection of an event in the environment or something more than that.

The rest of the data presented in the manuscript would all remain logical and fully consistent even if the activity interpretation is completely changed. For example, presence of resting state correspondences across groups and their relationship with stimulus evoke activity signal size as well as the relationship with white matter properties all point to the concept that evoked activity is grounded on brain resting state properties and white matter development. None of these results however provide any additional evidence that the observed signal is in fact related to nociception, and thus has any predictive value of pain perception in neonates.

We believe that our new analyses show clearly that the brain activity we observed expressed activity associated with pain – see our Summary of major changes to manuscript. Our results give us some understanding for what we'd expect to see in the control conditions outlined here. We expect that a gentle touch condition would produce a

response with some overlap with what we see, but we would expect these responses would be weaker²⁷, and would not reliably express the NPS signature, which has been shown to be specific to pain and correlates with the variability we see in our noxious-evoked response across subjects. We would not expect the mother's voice or other auditory stimuli to show significant spatial overlaps with our responses, and it is not possible in the current study to determine whether these responses could be predicted from resting-state data. We have been careful to make sure we are not making strong claims about an ability to predict pain perception. We argue that our ability to predict noxious-evoked responses, which includes processes related to nociception and pain, is in itself of substantial importance as it may begin to elucidate the neurophysiological bases for clinical infant pain scales, such as the PIPP-R score derived from behavioural and autonomic responses.

Simple biological correlates may have strengthened their argument but none is provided. The methods section mentions that heart rate and blood oxygenation was collected. One would be curious if these were correlated to the brain stimulus-response magnitude. Similarly, was there any evidence of reflexive responses to the stimulus and did these relate to brain evoked responses.

Unfortunately, heart rate and blood oxygen saturation were recorded, but were not of sufficient quality for analysis due to the minimal effect size due to the intensity of the pinprick stimulus. And while mild reflex activity was qualitatively observed, as previously reported⁸, quantitatively assessing reflex activity using EMG in the MRI environment was not possible due to clinical safety concerns. **Importantly, these clinical behavioural and autonomic/biological response measures cannot demonstrate the nociceptive nature of the noxious-evoked brain activity. They are neither specific to pain nor nociception, nor other non-nociceptive components of pain.** We now report previous observations published from our lab that provide some evidence for the relationship of these measures to our noxious-evoked measurements, and highlight that behavioural and autonomic responses also lack specificity to pain.

“Several infant clinical pain scales are multidimensional in nature²⁵, such as the PIPP-R scale which takes into account behavioural and autonomic responses. Previous results from our lab demonstrate that pinprick-evoked brain responses measured with EEG (electroencephalography) correlate with both reflex behaviour assessed with EMG (electromyography)¹ and autonomic heart rate responses⁵. However, technical and safety challenges in incorporating EMG into the MRI environment and unreliable estimates of inter-subject variability in autonomic responses (due to the small effect size associated with the pinprick stimulus), precluded linking the BOLD response amplitudes to these behavioural and autonomic dimensions at the individual level in the current study. But it is important to note that these behavioural and autonomic responses lack specificity to pain dimensions^{25,26}, and thus cannot be used to decompose the noxious-evoked BOLD response into sensory-discriminative, cognitive, or emotional contributions, or demonstrate that evoked BOLD responses differentially reflect nociception.” – page 16 paragraph 2.

We believe careful analysis using fMRI signature templates with validated associations to different experimental conditions and pain reports provides important supporting evidence for the processes contributing to the noxious-evoked response.

“Here, we adopted a template-based approach to demonstrate that variability in noxious-evoked activity in neonates shows concordance with adult fMRI activity associated with pain and nociception, and demonstrate the expression of the NPS, which tracks adult verbal pain ratings¹⁵. These results provide important supporting evidence for the processes contributing to the noxious-evoked response, and gives us some confidence that the brain responses reflect a set of pain-related processes with some concordance with existing infant and adult pain measures.” – page 16 paragraph 3.

The 3 cases where brain activity was negative raises many questions. The authors interpret them as a hemodynamic decoupling but this is pure speculation.

There are several mechanisms by which negative BOLD effects can be seen in adults, as recently reviewed²⁸: initial dip, reduced neural activity, blood-steal effect, subcortical endogenous neurotransmission, CSF volume changes, and increased metabolism with insufficient vascular response. While we cannot directly confirm our hypothesised interpretation that the noxious-evoked negative BOLD in primary sensory cortical regions reflect

haemodynamic decoupling, we believe it is the most plausible interpretation based on existing literature. We have now toned down our interpretation of this modest effect observed in a small number of subjects:

“In this maturity framework, our observed negligible or negative BOLD responses could be explained by immaturity in neurovascular coupling mechanisms, such as vasoconstriction or insufficient functional hyperaemia to meet oxygen demands²⁹. However, there are several mechanisms by which negative BOLD effects can be seen in adults²⁸. As the 3 negative BOLD responses we observed were generally of low magnitude, they may simply reflect effects of acquisition noise on a limited BOLD response.” – page 18 paragraph 3.

Overall, the general topic of nociception, pain, salience and their interrelationships, even in the adult, remains a complicated and contentious topic. Conclusions regarding pain in the neonate would require far better controlled studies.

As should be clear from our responses and updates to the text, we are in full agreement that disentangling nociception, pain, salience and their interrelationships is a difficult and complex topic, and agree that control conditions are an important element to achieving this. The changes to our manuscript emphasise this challenge, while identifying the value of characterising noxious-evoked brain responses and their relationship with brain structure and activity.

Reviewer #2 Comments (blue) and author reply (black):

The authors answered the majority of my questions. I have only minor comments to make.

- To give an comprehensive overview of the study, an additional figure could show a flow chart of the successive analysis steps (described in the results section), on the different groups of subjects (with their respective number) and based on the different MRI modalities (nociception paradigm fMRI, resting-state fMRI, diffusion MRI). This would really help to understand the analyses performed, and how they fit together (e.g. for the dHCP database, resting-state networks have been first estimated on N=242 newborns, while only N=215 are included in the correlation analyses with the diffusion MRI).

We are grateful to the Reviewer for continuing to highlight this source of confusion in our paper, and we agree that an additional figure summarising our study’s workflow would be helpful. We have included the listed information in this novel figure (Supplementary Figure 10), and reference the figure throughout our methods description.

- P26: The authors could specify the duration of the total acquisition protocol, as well as the duration for the fMRI protocol with the nociception paradigm, and for the diffusion MRI protocol.

This information is now included in the relevant methods section.

- Table 2: is the order of the subjects the same as in Figure 1? That would be relevant and to be specified.

Yes, the subject orders are the same. This relevant point is now included in the Table 2 caption.

- Supplementary Figure 6: For the histograms of response amplitude, wouldn't it be n=18 for the local dataset?

Yes, that is correct. This sample size information in the figure caption now correctly reads “n=18 for response amplitude histogram; n=17 for microstructure histograms”.

Supplementary Figure 10: Study workflow. Sequence of key steps (1-7), each step summarised using five details (a-e), for the current study. Note in step 3, the dHCP dataset ($n=242$) analysis and results were generated prior to and independent of the current study in a previous dHCP research output³⁰.

Abbreviations: a = dataset; b = imaging modality; c = key analysis method; d = sample size; e = key results figures and tables; dHCP = developing human connectome project; fMRI = functional MRI; dMRI = diffusion MRI; rs-fMRI = resting-state fMRI; RSN = resting-state network; MAD = median absolute deviation; PFM = probabilistic functional mode analysis.

Reviewer #3 Comments (blue) and author reply (black):

The authors have thoroughly addressed my previous comments and concerns in the rebuttal letter and the revised version of the manuscript. I have no additional comments.

We are glad to hear we have satisfactorily addressed the Reviewer's comments and concerns, and thank the Reviewer for their valuable contributions to improving this paper.

References:

1. Hartley, C. *et al.* The relationship between nociceptive brain activity, spinal reflex withdrawal and behaviour in newborn infants. *Sci. Rep.* **5**, 12519 (2015).
2. Johnston, C. C. *et al.* Factors explaining lack of response to heel stick in preterm newborns. *J. Obstet. Gynecol. Neonatal Nurs. JOGNN* **28**, 587–594 (1999).
3. Ranger, M., Johnston, C. C. & Anand, K. J. S. Current controversies regarding pain assessment in neonates. *Semin. Perinatol.* **31**, 283–288 (2007).
4. Verriotis, M. *et al.* Mapping Cortical Responses to Somatosensory Stimuli in Human Infants with Simultaneous Near-Infrared Spectroscopy and Event-Related Potential Recording. *eNeuro* **3**, (2016).
5. Hartley, C. *et al.* Nociceptive brain activity as a measure of analgesic efficacy in infants. *Sci. Transl. Med.* **9**, (2017).
6. Verriotis, M. *et al.* Cortical activity evoked by inoculation needle prick in infants up to one-year old. *Pain* **156**, 222–230 (2015).
7. Magerl, W., Fuchs, P. N., Meyer, R. A. & Treede, R. D. Roles of capsaicin-insensitive nociceptors in cutaneous pain and secondary hyperalgesia. *Brain J. Neurol.* **124**, 1754–1764 (2001).
8. Goksan, S. *et al.* fMRI reveals neural activity overlap between adult and infant pain. *eLife* **4**, (2015).
9. Iannetti, G. D., Baumgärtner, U., Tracey, I., Treede, R. D. & Magerl, W. Pinprick-evoked brain potentials: a novel tool to assess central sensitization of nociceptive pathways in humans. *J. Neurophysiol.* **110**, 1107–1116 (2013).
10. Slater, R. *et al.* Evoked potentials generated by noxious stimulation in the human infant brain. *Eur. J. Pain Lond. Engl.* **14**, 321–326 (2010).
11. Slater, R. *et al.* Cortical pain responses in human infants. *J. Neurosci. Off. J. Soc. Neurosci.* **26**, 3662–3666 (2006).
12. IASP. IASP Terminology - IASP. <https://www.iasp-pain.org/Education/Content.aspx?ItemNumber=1698> (2020).
13. Tracey, I. & Mantyh, P. W. The cerebral signature for pain perception and its modulation. *Neuron* **55**, 377–391 (2007).
14. Wiech, K. Deconstructing the sensation of pain: The influence of cognitive processes on pain perception. *Science* **354**, 584–587 (2016).
15. Duff, E. *et al.* Inferring pain experience in infants using quantitative whole-brain functional MRI signatures: a cross-sectional, observational study. *Lancet Digit. Health* **2**, e458–e467 (2020).
16. Goksan, S. *et al.* The influence of the descending pain modulatory system on infant pain-related brain activity. *eLife* **7**, (2018).
17. Gibbins, S. *et al.* Validation of the Premature Infant Pain Profile-Revised (PIPP-R). *Early Hum. Dev.* **90**, 189–193 (2014).
18. Stevens, B., Johnston, C., Taddio, A., Gibbins, S. & Yamada, J. The premature infant pain profile: evaluation 13 years after development. *Clin. J. Pain* **26**, 813–830 (2010).
19. Stevens, B. J. *et al.* The premature infant pain profile-revised (PIPP-R): initial validation and feasibility. *Clin. J. Pain* **30**, 238–243 (2014).
20. McDowell, I. The Theoretical and Technical Foundations of Health Measurement. in *Measuring Health: A guide to rating scales and questionnaires* (Oxford University Press, 2006).
21. Worley, A., Fabrizi, L., Boyd, S. & Slater, R. Multi-modal pain measurements in infants. *J. Neurosci. Methods* **205**, 252–257 (2012).
22. Vaart, M. *et al.* Multimodal pain assessment improves discrimination between noxious and non-noxious stimuli in infants. *Paediatr. Neonatal Pain* **1**, 21–30 (2019).
23. Yarkoni, T., Poldrack, R. A., Nichols, T. E., Van Essen, D. C. & Wager, T. D. Large-scale automated synthesis of human functional neuroimaging data. *Nat. Methods* **8**, 665–670 (2011).
24. Wager, T. D. *et al.* An fMRI-based neurologic signature of physical pain. *N. Engl. J. Med.* **368**, 1388–1397 (2013).
25. Lee, G. & Stevens, B. Neonatal and infant pain assessment. in *Oxford Textbook of Paediatric Pain* 353–369 (Oxford University Press, 2013).
26. Slater, R., Cantarella, A., Franck, L., Meek, J. & Fitzgerald, M. How well do clinical pain assessment tools reflect pain in infants? *PLoS Med.* **5**, e129 (2008).
27. Williams, G. *et al.* Functional magnetic resonance imaging can be used to explore tactile and nociceptive processing in the infant brain. *Acta Paediatr.* **104**, 158–166 (2015).
28. Goense, J., Bohraus, Y. & Logothetis, N. K. fMRI at High Spatial Resolution: Implications for BOLD-Models. *Front. Comput. Neurosci.* **10**, (2016).
29. Kozberg, M. & Hillman, E. Neurovascular coupling and energy metabolism in the developing brain. *Prog. Brain Res.* **225**, 213–242 (2016).
30. Fitzgibbon, S. *et al.* The developing Human Connectome Project automated functional processing framework for neonates. *OHBM* (2018).

Reviewer #1 (Remarks to the Author):

This manuscript has substantially improved. I like the new analyses as they certainly add additional evidence for observed activity being related to nociceptive processes. The authors also properly and thoroughly revised the terminology and make the caveats of the study more transparent.

Reviewer #2 (Remarks to the Author):

The authors have answered all my questions. The analysis that has been added seems to me relevant to support the physiological significance of the brain responses observed in newborns following the pinprick stimuli. I have only minor comments concerning the presentation of figures and tables added in the new version of the manuscript.

- Figure 2bi: the legend might be improved to clarify the correspondence with the different functional templates in x axis. Reordering them in x axis might be useful. Is "Social Rejection Pain" the fourth or the last point of the plot as it is supposed to be a "pain signature negative control"?
- Figure 2bii-iii: the link with Supplementary Figure 9 might be mentioned. Statistics of linear regressions might be added in the plots. It might be interesting to show the plots for all templates showing significance in 2bi.
- Supplementary Figure 9: Statistics of linear regressions might be added in the plots. Regression lines might be removed for non-significant tests.
- Keeping Table 3 in first position (Table 1) would make sense.
- Table 1 (becoming Table 2?) might refer to Figure 2 and Sup Figure 9.
- Tables 1 and 2: Have corrections for multiple comparisons been performed?

Reviewer #1 Comments (blue) and author reply (black):

This manuscript has substantially improved. I like the new analyses as they certainly add additional evidence for observed activity being related to nociceptive processes. The authors also properly and thoroughly revised the terminology and make the caveats of the study more transparent.

We would like to thank the reviewer for their very helpful comments throughout the review process, which have significantly improved our manuscript.

Reviewer #2 Comments (blue) and author reply (black):

The authors have answered all my questions. The analysis that has been added seems to me relevant to support the physiological significance of the brain responses observed in newborns following the pinprick stimuli. I have only minor comments concerning the presentation of figures and tables added in the new version of the manuscript.

- Figure 2bi: the legend might be improved to clarify the correspondence with the different functional templates in x axis. Reordering them in x axis might be useful. Is “Social Rejection Pain” the fourth or the last point of the plot as it is supposed to be a “pain signature negative control”?

The “Social Rejection Pain” is the last point of the plot. Similar to “Visual”, “Social” is a negative control. The x-axis is ordered based on the nature of the template origin rather than its purpose i.e. data-derived template first (Noxious-evoked), followed by the seven Neurosynth templates (Visual to Arousal), followed by the Pain subtype signature templates (NPS and Social). Additional text has been to the figure legend to clarify this point on x-axis order.

Figure 2bii-iii: the link with Supplementary Figure 9 might be mentioned. Statistics of linear regressions might be added in the plots. It might be interesting to show the plots for all templates showing significance in 2bi.

The link with Supplementary Figure 9 is now mentioned, and the statistical results have been added to the figure caption. Due to all Neurosynth and pain signature correlation plots being presented in Supplementary Figure 9, we have only highlighted the two pain templates (Neurosynth Pain and NPS) in the main text, as the primary focus is on demonstrating the relationship between overall noxious-evoked responses and pain responses.

- Supplementary Figure 9: Statistics of linear regressions might be added in the plots. Regression lines might be removed for non-significant tests.

In line with the style of other figures in our manuscript, the statistics have been added above each plot, and the regression lines have been included for both significant and non-significant associations. A reference to Table 1 has also been added.

- Keeping Table 3 in first position (Table 1) would make sense.

Table 3 has now been removed based on editorial feedback.

- Table 1 (becoming Table 2?) might refer to Figure 2 and Sup Figure 9.

Table 1 now refers to these two figures.

- Tables 1 and 2: Have corrections for multiple comparisons been performed?

The reported p-values for both tables are uncorrected for multiple testing. Compared to reporting adjusted p-values, we believe that reporting the unadjusted p-values will give the reader a better perspective on the individual strength of each effect. However, we fully agree that the reported statistical significance must survive thresholding that includes corrections for multiple testing. To achieve both these aims, we now report the adjusted significance level (rather than the adjusted p-values), which includes an adjustment for multiple testing. In both tables, the

significance level has been Bonferroni-corrected for effective number of tests^{1,2}, which appropriately takes into account the non-independence of the tests due to correlations among the response variables. The introduction of corrections for multiple comparisons does not affect the inferences in our paper.

Note that the p-values in Table 2 have been updated. Previously, we had only used 1,000 permutations to generate the p-values for the prediction results, which did not provide enough granularity (decimal points) to judge significance relative to the adjusted significance level. We have now generated all p-values using 10,000 permutations.

We would like to thank the reviewer for their detailed, thorough, and thoughtful feedback throughout this review process. This input has substantially improved the quality of our manuscript.

References

1. Derringer, J. A simple correction for non-independent tests. (2018) doi:10.31234/osf.io/f2tyw.
2. Nyholt, D. R. A Simple Correction for Multiple Testing for Single-Nucleotide Polymorphisms in Linkage Disequilibrium with Each Other. *Am. J. Hum. Genet.* **74**, 765–769 (2004).